# PLANSFORMER: GENERATING SYMBOLIC PLANS USING TRANSFORMERS

## ABSTRACT

Large Language Models (LLMs) have been the subject of active research, significantly advancing the field of Natural Language Processing (NLP). From BERT to BLOOM, LLMs have surpassed state-of-the-art results in various natural language tasks such as question answering, summarization, and text generation. Many ongoing efforts focus on understanding LLMs' capabilities, including their knowledge of the world, syntax, and semantics. However, extending the textual prowess of LLMs to symbolic reasoning has been slow and predominantly focused on tackling problems related to the mathematical field. In this paper, we explore the use of LLMs for automated planning - a branch of AI concerned with the realization of action sequences (plans) to achieve a goal, typically executed by intelligent agents, autonomous robots, and unmanned vehicles. We introduce Plansformer[1]; an LLM fine-tuned on planning problems and capable of generating plans with favorable behavior in terms of correctness and length with reduced knowledge-engineering efforts. We also demonstrate the adaptability of Plansformer in solving different planning domains with varying complexities, owing to the transfer learning abilities of LLMs. For one configuration of Plansformer, we achieve  97% valid plans, out of which  95% are optimal for Towers of Hanoi - a puzzle-solving domain.

## 1 INTRODUCTION

Large Language Models (LLMs), based on transformer-based (neural) architecture (Vaswani et al., 2017; Devlin et al., 2018; Brown et al., 2020; Chowdhery et al., 2022), have significantly advanced the field of Natural Language Processing (NLP). Their employment has grown dramatically in recent times (Li, 2022), as researchers develop newer and bigger LLMs. From BERT to the most recent BLOOM, language models have surpassed state-of-the-art results in various natural language tasks. For example, PaLM (Chowdhery et al., 2022), achieved breakthrough performance on a plethora of natural language tasks such as inference, question answering, and commonsense reasoning, and outperformed an average human performance on the BIG-bench benchmark.

Despite the textual prowess of LLMs, their significance has been limited in the domains that involve symbols. For example, domains with symbols such as mathematics (Hendrycks et al., 2021b; Cobbe et al., 2021), and coding problems (Hendrycks et al., 2021a; Chen et al., 2021) deliberates the failures of LLMs when it comes to handling symbols. In Automated Planning, (Valmeekam et al., 2022) suggests that even state-of-the-art LLMs cannot reason with symbolic data and offer a new suite of benchmarks to test their reasoning capabilities. Recently, there has been a lot of interest in LLMs for code generation; for example, CodeT5 (Wang et al., 2021), CodeBERT (Feng et al., 2020), Codex (Chen et al., 2021), etc. In this paper, we propose to employ LLMs that are trained to generate code and repurpose them to generate valid plans. To advance the research in LLM-based automated planning, we create a training and test dataset for several planning domains. We use CodeT5 (base), a transformer-based code generation model that achieves state-of-the-art results in CodeXGlue, as the pre-trained LLM. We select CodeT5 due to its ability to generate goal-directed, sequential instruction and semantically meaningful program codes with syntactic and structural constraints. Then, we present, Plansformer, an LLM trained to generate symbolic plans of high quality in terms of correctness and length. Our experimental results indicate that the syntactic/symbolic knowledge learned from different programming languages in the CodeT5 model can be beneficial for the PDDL-based automated planning task. For example, in one configuration of Plansformer tested on a puzzle-solving domain, Towers of Hanoi - **hanoi**, our model was able to generate 97%

---

[1]The dataset, code, and all fine-tuned model checkpoints will be released for public usage.

valid plans, out of which 95% are shortest length plans. The results reveal a promising direction to harness LLMs for symbolic tasks such as planning.

In the remainder of the paper, we present preliminaries on automated planning and language models and then propose an LLM repurposed planner called *Plansformer*. Next, we present the experimental results comparing our approach with state-of-the-art planners and other large language models. Furthermore, we demonstrate the ability of Plansformer to adapt to other domains and discuss the relevance to instruction generation. We conclude with a discussion of the results and presentation of ongoing work.

## 2 BACKGROUND

### 2.1 AUTOMATED PLANNING

Given the initial and goal states, alongside a set of legal actions, the objective of a planning agent is to devise a sequence of actions that advance the agent from the initial to the goal state. This paper adopts the Planning Domain Description Language (PDDL) (McDermott et al., 1998; Fox & Long, 2003) notations. In PDDL, a planning environment is described in terms of objects in the world, predicates that describe relations between these objects, and actions that modify the world by manipulating these relations. The output plan consists of a series of time steps, each of which can have one or more instantiated actions with concurrency semantics (Ghallab et al., 2004). A planner devises plans by searching in the space of states, where a state is a configuration of physical objects or partial plans. There is a single agent in the most basic formulation, called classical planning. The actions have unit cost, take constant time to execute, have deterministic effects, with the fully observable world, domain-specific conditions/constraints, and all goals have to be achieved (Ghallab et al., 2004). In more sophisticated planning settings, many of these conditions are relaxed. There may be multiple agents, and the cost and duration of actions can be non-uniform. At the same time, its effects can be non-deterministic, the world can be partially observable, and the agent may maximize as many goals as it can achieve in a given time and resource budget.

### 2.2 LARGE LANGUAGE MODELS AND SYMBOLIC TASKS

Large Language Models (LLM) such as BERT (Wolf et al., 2020), RoBERTa (Liu et al., 2019), and GPT3 (Brown et al., 2020) are pre-trained on extensive unstructured knowledge from public data such as Wikipedia, Bookcorpus, and Commoncrawl, have shown impressive results in several NLP tasks. It demonstrated the ability to generalize to multiple tasks from question answering and machine translation to story generation and instruction following (Wang et al., 2018; 2019). LLMs have shown the ability to generate output in natural language (Wolf et al., 2020; Raffel et al., 2020), adapt to novel tasks in a zero or few-shot approach (Brown et al., 2020; Radford et al., 2019) and decode with constraints on output space (Hokamp & Liu, 2017; Welleck et al., 2019; Kumar et al., 2021). Recent progress in LLMs has demonstrated the generation of structured output that requires precise syntactic/symbolic knowledge with structural constraints such as knowledge graphs (Petroni et al., 2019), protein structure (Unsal et al., 2022; Ferruz & Höcker, 2022), and programming languages (Ahmad et al., 2021). As the LLMs collect the related knowledge necessary to solve an NLP task, (Petroni et al., 2019) have shown that the LLMs are potential representations of the significant knowledge bases. In protein data (Unsal et al., 2022; Ferruz & Höcker, 2022), LLMs generate the functional properties of the proteins by enforcing the structural constraints specific to protein science and determining the complex functional relationships in the protein binding. Code generation has recently become very popular in the LLM research community. Several models such as CodeBERT (Feng et al., 2020), Codex (Chen et al., 2021), and CodeT5 (Wang et al., 2021) have shown significant improvement in transfer from pre-trained models for natural language to structured codes. One of the key contributors to the success of these LLMs in code generation is fine-tuning the models on task-specific data. For instance, CodeXGlue (Lu et al., 2021), a benchmark dataset for code understanding and generation with sample codes from several programming languages, is used to fine-tune CodeBERT, CodeT5, and others. In this paper, we harness CodeT5 for further fine-tuning to the classical automated planning domain due to its ability to generate goal-directed, sequential instruction and semantically meaningful program codes with syntactic and structural constraints.

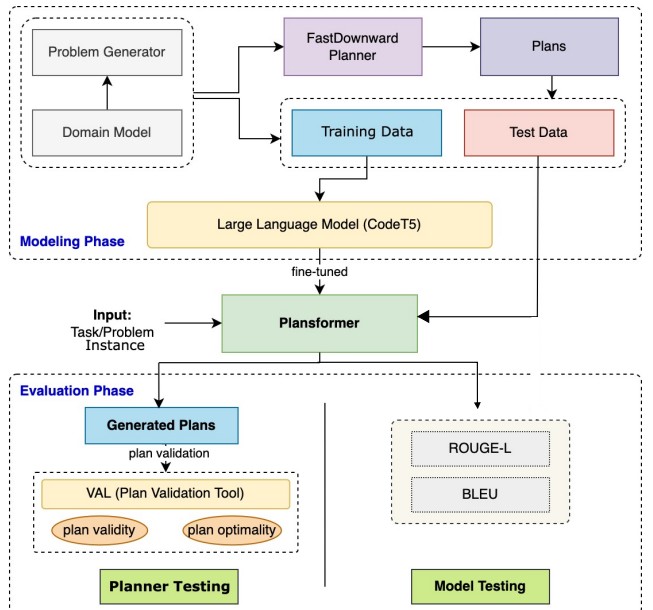

Figure 1: Plansformer Model Architecture showing modeling and evaluation phases. Modeling phase involves finetuning CodeT5 with data from planning domain. Evaluation phase shows both the planner and model testing.

The closest prior art addressing the ability of LLMs to generate symbolic plans are the studies of (Hernandez et al., 2021; Valmeekam et al., 2022; Huang et al., 2022). (Hernandez et al., 2021) looks at different ways to generate plans using GPT-3 versions, a prominent generative LLM. (Valmeekam et al., 2022) discusses different scenarios for generating plans, like finding a satisficing plan or adapting a previous one, and discusses encodings to generate plans and verify using a plan validator. (Huang et al., 2022) generates step-by-step instructions for a user-defined task using LLM prompting. All these studies require the (human-guided) mapping of a natural language-based sequence of instructions generated by the LLM to the admissible action in the planning domain as an additional step.

## 3 PLANSFORMER FOR SYMBOLIC PLANS

Figure 1 provides an illustrative overview of how we generate and test our planner, called Plansformer. *The first phase, modeling, shows how we fine-tune the CodeT5 to address planning syntax and semantics.* The second phase, *evaluation*, deals with assessing the competency of Plansformer as a model and as a planner. The key idea here is to utilize an LLM (CodeT5) pretrained on code generation and further train it on planning problem instances with corresponding valid plans. We evaluate its competence in generating valid plans (or almost valid plans for unseen planning problem instances) using two types of testing: 1) Model testing measures if Plansformer could generate meaningful responses (as in the test dataset), 2) Planner testing measures if the generated plans are valid/optimal (independently of whether they were the same plans as in the test dataset).

### 3.1 MODELING PHASE

In the modeling phase, we first create a planning-based dataset for finetuning the CodeT5 to generate plans. The modeling phase of Figure 1 depicts the different modules employed.

#### 3.1.1 PLANNING DATASET

We generate a PDDL-based dataset[2] as a benchmark to finetune pretrained CodeT5 and facilitate further research at the intersection of LLMs and automated planning. We use the domain model

---

[2]The link is omitted during the review process.

| Task | Problem Instance | Plan |
|------|-----------------|------|
| blocksworld | **\<GOAL\>** on b1 b2, on b2 b3, ontable b3, on b4 b1, clear b4
**\<INIT\>** handempty, ontable b1, clear b1, on b2 b3, ontable b3, on b4 b2, clear b4
**\<ACTION\>** pick-up
   **\<PRE\>** clear x, ontable x, handempy
   **\<EFFECT\>** not ontable x, not clear x, not handempty, holding x
**\<ACTION\>** put-down
   **\<PRE\>** holding x
   **\<EFFECT\>** not holding x, clear x, handempty, ontable x
**\<ACTION\>** stack
   **\<PRE\>** holding x, clear y
   **\<EFFECT\>** not holding x, not clear y, clear x, handempty, on x y
**\<ACTION\>** unstack
   **\<PRE\>** on x y, clear x, handempty
   **\<EFFECT\>** holding x, clear y, not clear x, not handempty, not on x y | unstack b4 b2, put-down b4, pick-up b1, stack b1 b2, pick-up b4, stack b4 b1 |

Figure 2: Snapshot of one instance of the plan dataset for blocksworld domain.

(in PDDL) to generate corresponding valid problem files with varying complexities automatically. In this paper, we focus on four different classical planning domains, i.e., *Blocksworld, Towers of Hanoi, Grippers, Driverlog.*

*Blocksworld*, or **bw**, is a well-studied domain (Gupta & Nau, 1991) with blocks placed on a table or arranged in vertical stacks. Here, one can alter the arrangement of the blocks with the available actions such as pick-up, put-down, stack, and unstack. We generate the problems with 2 to 5 block configurations.

*Towers of Hanoi*, or **hn**, consists of 3 pegs and multiple disks of varying diameters. Initially, all disks are placed in the first peg, and the end goal is to move the disks to the last peg. The only limitation to consider when moving the disks is that only a smaller disk can be placed on top of a bigger disk. Although the domain has only one action, the problem-solving is recursive (Gerety & Cull, 1986). Here, we generate the problems with configurations of 2 to 5 disks.

*Grippers*, or **gr** domain involves moving balls across rooms using robotic grippers. It has problems generated with configurations of 2 to 5 balls, 3 to 5 robots, and 2 to 4 rooms.

*Driverlog* or **dl** domain involves moving packages on trucks between locations driven by drivers. It has problems generated with configurations of 1 to 3 drivers, 1 to 3 trucks, 2 to 4 packages, and 3 to 6 locations.

Each planning domain explained above includes multiple problem instances. We generate the corresponding plans for each problem instance using FastDownward planner (Helmert, 2006). FastDownward is a classical planning system based on a heuristic search and offers different search algorithms such as causal graph heuristics and A$^*$ search. FastDownward can generate optimal plans with A$^*$ LM-Cut heuristic (Helmert & Domshlak, 2011). Hence, FastDownward can be regarded as a potent planner for generating a dataset of optimal plans.

We show the snapshot of the generated dataset in Figure 2, with more examples in Section 1 of supplementary material. Unlike the traditional planner which requires two different files - domain and problem (as pddl files, See Figure 2 in supplementary material), Plansformer reduces the knowledge-engineering efforts with a simplified input format that includes the problem instance coupled with a corresponding valid plan. The problem instance captures all the essential information in the domain and problem instance, such as the goal, the initial state, and the possible actions that can be taken in that domain. The generated dataset for each domain consists of 18,000 plans with different problem configurations. For training, we use 5-fold cross-validation with an 80%-20% split of the generated dataset for each domain. The average plan length (number of actions in the generated plan) for blocksworld is 9, gripper is 9, driverlog is 10, and hanoi is 12.

### 3.1.2 TOKENIZER

We use a Byte-level BPE tokenizer, following the standard practice in LLMs, with a vocabulary size of 32,005. We add PDDL-specific tokens, namely, `[GOAL]`, `[INIT]`, `[ACTION]`, `[PRE]`, `[EFFECT]` to represent the goal state, initial state, possible actions with their associated preconditions and effects these actions cause in the environment respectively. We do not re-train a specific tokenizer for this task from scratch following the previous work (Chen et al., 2021), where GPT-3's tokenizer was reused to generate code.

### 3.1.3 Fine-tuning CodeT5

While there are many LLMs to select as a candidate for this work, we shortlist the models pre-trained on code generation to exploit the syntactic information in the programming languages implicitly captured in their weights. Although Codex (Chen et al., 2021), built using GPT-3, has reported the best performance in solving code-related tasks, its lack of public access led us to choose an equally competitive LLM: CodeT5 (Wang et al., 2021). CodeT5 is a masked language model consisting of an encoder-decoder stack inspired by the transformer architecture (Vaswani et al., 2017). It is capable of performing a wide range of tasks including code generation and understanding tasks. The generation tasks include code summarization, code generation, translation, and refinement. The understanding tasks include code defect detection and clone detection. CodeT5 is pretrained with example codes from eight programming languages - Python, Java, JavaScript, PHP, Ruby, Go, C, and C#. Its pre-training tasks include identifier awareness and bimodal generation, which optimizes code-to-code understanding. The CodeT5 model possesses several properties amenable to the planning domain, such as its ability to generate goal-directed, sequential instruction and semantically meaningful program codes with syntactic and structural constraints. With this pre-trained knowledge already encoded within CodeT5, we finetune it with $14400$ samples ($80\%$ of the generated dataset) for each independent domain from the planning dataset. As a result of this finetuning, the weights of CodeT5 are updated to account for the task of plan generation. We give the planning problem instance as input to CodeT5's encoder and generate the intermediate features for the decoder of CodeT5 to output a plan.

## 3.2 Evaluation Phase

Plansformer is an LLM that ingests a new problem instance as input and outputs a plan for that problem instance. Therefore, to evaluate its competency, we must test its quality as a model and planner. The evaluation phase is described in the lower part of Figure 1, showing both testing phases.

### 3.2.1 Planner Testing

Unlike in natural language, symbolic plans have richer information content, inherently captured in their structure. Thus, we have an additional evaluation phase for plan validation to check how well Plansformer can mimic an automated planner. The sequence of actions generated by Plansformer must help an agent to navigate from the initial state to the goal state for a given problem instance. We call a generated plan *cost-optimal* [3] if it is the shortest possible among all other plans. Several metrics exist in the automated planning literature to evaluate a plan generated by Plansformer. In this paper, we consider *validity* and *optimality*. We evaluate the plan generated by Plansformer using a plan validation tool, called VAL (Howey et al., 2004), to check for its optimality and validity. VAL is an automatic validation tool for PDDL. VAL takes as input the task posed to Plansformer and the corresponding generated plan. It applies PDDL-based relaxation conditions to check for validity and optimality.

### 3.2.2 Model Testing.

It is typical to evaluate natural language tasks such as summarization or generation using metrics such as BLEU and ROUGE. Both BLEU and ROUGE are widely used metrics in NLP. In general, BLEU measures precision and helps understand how closely a machine translation (here, plan generated by Plansformer) is compared to a human translation (here, plan generated by an automated planner). On the other hand, ROUGE measures recall, i.e., how many of the words referenced in human summaries appeared in the summaries generated by the machine. In particular, we adopt ROUGE-L, which considers sentence-level structure similarity by identifying the longest co-occurring sequence n-grams. Although ROUGE and BLEU have no direct intuition in automated planning, we use these metrics to look at the task of plan generation from the perspective of LLMs. The evaluation based on these metrics provides us with an insight into the performance of Plansformer as a language model. In the next section, we evaluate Plansformer as a planner to give conclusive evidence on how well Plansformer generates the plans.

---

[3]We also refer to it as optimality interchangeably

| Models | ROUGE-L$_{\text{recall}}$ | ROUGE-L$_{\text{precision}}$ | ROUGE-L$_{\text{fmeasure}}$ | BLEU |
|---|---|---|---|---|
| Codex | 0.72 | 0.52 | 0.60 | 0.36 |
| GPT-2 | 0.04 | 0.14 | 0.06 | 0.07 |
| T5-base | 0.16 | 0.70 | 0.26 | 0.02 |
| CodeT5-base | 0.41 | 0.28 | 0.33 | 0.02 |
| Plansformer | **0.93** | **0.93** | **0.93** | **0.89** |
| Plansformer-bw | 0.97 | 0.99 | 0.98 | 0.90 |
| Plansformer-hn | 0.99 | 0.96 | 0.97 | 0.95 |
| Plansformer-gr | 0.94 | 0.94 | 0.94 | 0.92 |
| Plansformer-dl | 0.82 | 0.83 | 0.82 | 0.79 |

Table 1: Results of model testing (best performance in bold).

## 4 EXPERIMENTAL RESULTS

In this section, we present the quantitative and qualitative results obtained using Plansformer to generate symbolic plans for multiple domains of varying complexities. We select a test-bed of $3,600$ unique and unseen problem instances ($20\%$ of the dataset) for each domain for evaluating Plansformer. All the results reported in this paper are averaged over 5 randomly selected ($80\% - 20\%$) train-test splits. We report the results for the Plansformer variants by evaluating the corresponding test-bed. For example, `Plansformer-bw`'s results are reported based on the performance results obtained on **bw** test-bed. We evaluate Plansformer using both model and planner testing to find its efficiency as a *language model* and a *planner*.

### 4.1 IS PLANSFORMER A GOOD MODEL?

Plansformer has an encoder-decoder pair, where the encoder attends to tokens on either side of the masked word, whereas the decoder auto-regressively generates plans. Table 1 compares all the Plansformer models with other LLMs using the model evaluation metrics (ROUGE and BLEU). In this experiment, we consider the best-performing models from (bidirectional) masked language models (e.g., T5 (Raffel et al., 2020)) and (unidirectional) causal language models (e.g., GPT-2 (Radford et al., 2019)) for the experiments. We present the actual plan generations from a few of these models in Figure 5 of supplementary material.

We report the performance of the baseline models averaged over the four planning domains. We also show the performance of Plansformer on individual domains (Plansformer-bw, Plansformer-hn, Plansformer-gr and Plansformer-dl). We observe that Plansformer performs best on all metrics, followed by Codex, with a significant ROUGE-L$_{\text{recall}}$ score. We believe that the performance gain from Codex compared to other baseline models is due to it's ability to relate the natural language understanding (a skill inherited from GPT-3) with code generation. It is interesting to see that CodeT5 performs poorly compared to Codex and Plansformer, demonstrating the advantages of the natural language understanding with code generation task on this evalutation metrics. We conclude that the models pre-trained with code-related tasks have an advantage over other models in plan generation task due to the similarities of PDDL with other programming languages. Despite with the best model testing metrics, We need to test Plansformer for plan validation to see its effectiveness as a planner.

### 4.2 IS PLANSFORMER A GOOD PLANNER?

In this section, we report the results from the planner testing. We evaluate the generated plans for validity and optimality. We also report the average time taken to solve the problem instances. Table 2 shows the plan validation scores obtained by different models. We consider FastDownward (Helmert, 2006) for plan validation to generate the ground truth plans. FastDownward planner generates a $100\%$ valid and optimal plan for a given input (i.e., a combination of domain description and problem instance) when the landmark-cut heuristic is used within a standard A* search framework (Helmert & Domshlak, 2011).

We can see that Blockworld **bw** domain achieved the highest performance gain via `Plansformer-bw` - generated 90.04%, out of which 88.44% are optimal. This better performance is analogous to the fact that **bw** is the easiest domain among the four domains. Although it

| Models | Valid Plans (%) | Invalid Plans | | Optimal Plans (%) | Avg. Time (sec) |
|---|---|---|---|---|---|
| | | Incomplete/Wrong (%) | Failed (%) | | |
| FastDownward (Ground Truth) | 100% | - | - | 100% | 10.28s |
| Codex | 0.15% | 0% | 99.85% | 0.15% | 1s |
| GPT-2 | 0% | 100% | 0% | 0% | 0.05s |
| T5-base | 0.25% | 82.7% | 17.3% | 0.25% | 0.47s |
| CodeT5-base | 0.6% | 99.4% | 0% | 0.6% | 0.68s |
| Plansformer | **83.64%** | 16.18% | 0.19% | **73.27%** | **0.06s** |
| Plansformer-bw | 90.04% | 9.94% | 0.02% | 88.44% | 0.05s |
| Plansformer-hn | 84.97% | 14.72% | 0.31% | 82.58% | 0.05s |
| Plansformer-gr | 82.97% | 16.61% | 0.42% | 69.47% | 0.06s |
| Plansformer-dl | 76.56% | 23.44% | 0% | 52.61% | 0.09s |

Table 2: Results of plan validation.

is hard to find optimal plans, we can find a valid plan linear in the number of blocks to any problem instances by putting them down and picking from the table (Gupta & Nau, 1991).

On a relatively more complex domain, i.e., **dl**, `Plansformer-dl` achieves 76.56% valid plans, out of which 52.61% are optimal. We notice a  20% difference between valid and optimal plans for **dl**, with an observation that the model can come up with completely new and valid action sequences, although may not optimal. We can see that the number of optimal plans generated reduces with the increasing complexity of the domains. We include both incomplete/wrong generations from the models and failed plans when reporting invalid plans. An incomplete/wrong generation is a partially correct ordering of action sequences, whereas a failed plan is an entire plan consists of impossible ordering of actions, not allowed by the domain definition. Figure 3 shows an example of incomplete generation and failed plans. All Plansformer models generate close to 0% failed plans for respective domains. It is a testament to Plansformer models' ability to understand and generate valid action sequences for a given domain.

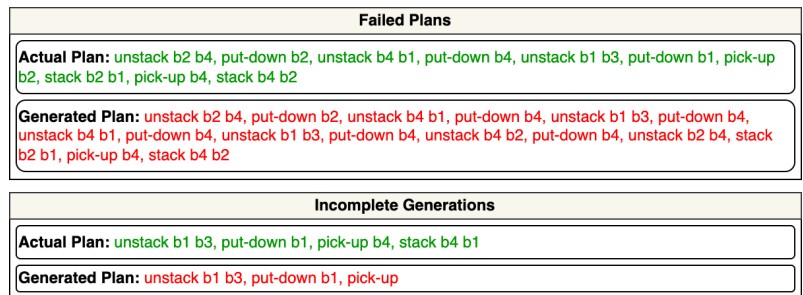

Figure 3: Different types of invalid plans generated by Plansformer on Blocksworld domain (Plansformer-bw).

Codex, the second best performing model according to ROUGE and BLEU scores, only generates 0.15% valid plans, emphasizing the need for a two-stage evaluation phase - where both model and generated plans are tested. We notice that the average time taken by Plansformer to completely solve the test-bed of problems is 2̃00x faster than the FastDownward, an automated planner that generated ground truth plans. Plansformer may offer an immense advantage in generating approximately correct plans in real-time applications. Interestingly, CodeT5, used to build Plansformer, takes considerable time to solve the same problem instances from the test bed. We believe that the Plansformer is faster since it generates valid and likely optimal plans shorter in length than usually long incoherent sequences generated by CodeT5, which tend to be time-consuming. The difference between plans generated by CodeT5 and Plansformer for the same problem is shown in Figure 4.

There have been very few relevant works that can be compared with Plansformer. Although some recent works such as (Huang et al., 2022) use LLMs to generate "plans", it is different from automated planning and is not symbolic in nature. These "plans" are step-by-step actions to perform a trivial everyday task such as *"Brush teeth"*. These methods use a user-constructed prompt to enable an LLM to generate appropriate steps from its prior knowledge. We believe the work by (Valmeekam et al., 2022) is similar in spirit to ours, using a PDDL-based natural language prompt to obtain symbolic plans using GPT-3. The significant difference between their dataset and ours is the difference in the object names, for example, a block is named as *b1* in our dataset as opposed

to *a* in (Valmeekam et al., 2022). This difference lets us evaluate Plansformer[4] on the dataset from (Valmeekam et al., 2022) with randomized object names as opposed to our dataset introduced in Section 3.1.1. On this dataset, Plansformer generated $66\%$ *valid plans*, whereas, GPT3 with PDDL-based natural language prompting generated only $0.6\%$ valid plans. The significant difference in performance enables us to validate the advantage of our approach in generating valid plans despite randomized object names.

Plansformer, trained and tested on the same domain, displays superior performance both as a model and a planner. However, LLMs are also well known for transfer learning, i.e., a model trained for solving one domain can be re-purposed to solve other related domains. In the next section, we explore how Plansformer trained on one domain can be adapted to another.

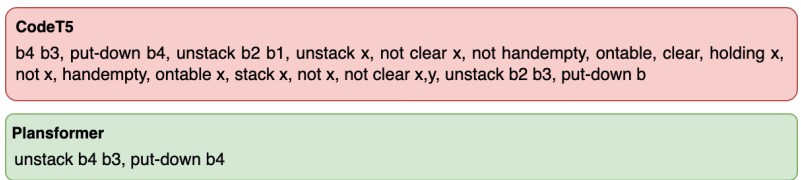

Figure 4: Long invalid generation from CodeT5 as opposed to a valid optimal plan from Plansformer-bw for the same problem.

## 4.3 CAN PLANSFORMER ADAPT TO ANOTHER DOMAIN?

The *base models*, i.e., Plansformer-x, where x can **bw, hn, gr, and dl**, cannot generate valid plans for other domains (i.e., `Plansformer-bw` on **hn, gr, and dl**) since each domain differs from the others in terms of action space, complexity, and solving. However, LLMs allow us to utilize the model trained in one domain to adapt to another using either making use of prompt conditioning or transfer learning with further fine-tuning on the problem instances from the new domain. We have seen from the previous work on prompt conditioning (Valmeekam et al., 2022) that the performance of the model on an unseen domain is very sensitive to the manually-identified prompt. A small perturbation to the prompt can significantly affect the model's performance, and creating a perfect prompt requires understanding the inner workings of LLM on hand and trial and error. In recent years, researchers have started looking at automatic prompt generation (Shin et al., 2020), which we would like to explore in the future.

Instead of the prompt conditioning, we follow the transfer learning approach by finetuning Plansformer *base models* with problem instances from other domains to check the ability of Plansformer to adapt to new domains. For brevity, we demonstrate variants of `Plansformer-bw` models on three other domains. Figure 5 shows different `Plansformer-bw` models and their plan validation scores on respective test-bed from target domains. We report that the results for transfer learning setup of Plansformer *base models* convey the same insights as `Plansformer-bw` shown here and are presented in Section 4.2 of supplementary material .

We consider different numbers of problem instances for finetuning `Plansformer-bw` on a given domain to see how the performance of the model varies across the sample size. We use the model naming format to convey the details on the amount of problem instances used for finetuning the Plansformer base model, i.e., `bw-hn[500]` implies that we further finetune `Plansformer-bw` using 500 problem instances from **hn** and report the results. In Figure 5, we can see an overall increase in the number of valid plans for every testing domain as we increase the problem instances available for finetuning. We observe that the models fine-tuned with 2000, which is $14\%$ of the training size of *base models*, achieves $50\%$ of the valid plans recorded by `Plansformer-hn`, `Plansformer-gr`, and `Plansformer-dl`. Despite the complexity of these planning domains, we obtain $> 90\%$ valid plans for all testing domains by increasing the finetuning samples to that of the training size of *base models*. `Plansformer-bw-hn[14400]` obtains the best performance among all models, by achieving 97.05% valid plans, out of which 95.22% are optimal. In Figure 5(d), we compare Plansformer-hn trained with different number of data points from **hn** domain

---

[4]We would like to note that Plansformer is trained only on the dataset introduced in Section 3.1.1 and is not trained/finetuned on the dataset in (Valmeekam et al., 2022) for this experiment

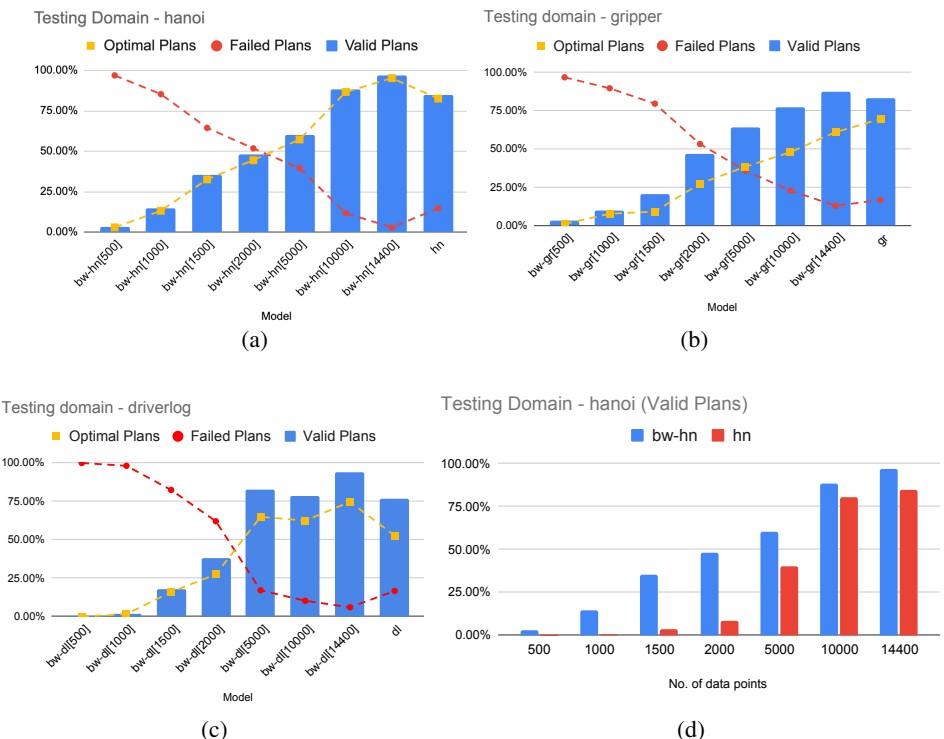

Figure 5: Plansformer-bw as the base model fine-tuned with and tested on (a) **hanoi** (b) **grippers** (c) **driverlog**, and (d) shows the comparison of valid plans generated by Plansformer-bw-hn derived models with Plansformer-hn trained using similar data points.

against the Plansformer-bw (base model) finetuned on the same data points. We can see a clear advantage of the transfer learning capability in LLMs, as both the **bw** and **hn** domains have similar plan semantics. Similar trends can be seen for the other domains and the results are reported in Section 4.2 of supplementary material.

We notice that the failed plans decrease with additional problem instances used for finetuning. Using the same amount of problem instances as training $(14, 400)$, we observe that the number of failed plans is less than that of the *base models* built for the respective domains. The number of optimal plans consistently increases with the number of problem instances in **hn** domain for finetuning. It is $13\%$ more than `Plansformer-hn`, whereas we can see some variations for the other two domains. We also report the fine-tuning results of all possible Plansformer models and their performance in Figure 6 of the supplementary material. We have also trained a single Plansformer model on all the domains (multi-task setting) and found that the individual model has relatively comparable performance to that of 5 (See supplementary material Section 4.2 for more details).

## 5 CONCLUSIONS AND ONGOING WORK

In this paper, we have explored using LLMs to generate symbolic plans for multiple domains. We have taken an LLM tailored to code and trained it further over a set of planning problem instances and corresponding valid plans. We then tested the model's capability to generate plans for unseen planning problem instances, evaluating the correctness and length of such plans. Our approach is compared to an existing state-of-the-art planner, showing that our LLM-based planner, called Plansformer, can solve most instances with high quality both in terms of correctness and length while needing *much less time* to generate such plans. Although Plansformer shows promising results, this work is not done and can be extended in many directions. One can seek to eliminate invalid plans comprising both failed or incomplete plans, drive efficiency further, explore more planning domains, e.g., from competitions (ICAPS, 2022), and tackle other planning settings (Valmeekam et al., 2022).

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
