# OpenReview forum: "Plansformer: Generating Multi-Domain Symbolic Plans using Transformers"
_ICLR.cc/2023/Conference — Submitted to ICLR 2023_

### Official Review · Reviewer_abxA · 2022-10-24

**Confidence:** 4
**Correctness:** 2
**Technical Novelty And Significance:** 2
**Empirical Novelty And Significance:** 2
**Recommendation:** 8

**Clarity, Quality, Novelty And Reproducibility:**

This paper is overall clear and straightforward on the approach used.  The reproducibility looks good with the dataset provided.
The experiment section has provided quantitative and qualitative results on both modeling metrics and plan validity metrics. It would be better if the data generation and representation can be illustrated clearer. The title and some claims in the paper may be revised as the models are mainly tested on one single domain at a time. Specifically, there is no experiment showing one model generating plans on multiple domains. Section 4.3 (and the supp material) suggests that the performance drops a lot if finetuned on another domain.
In terms of motivation, it would be helpful to discuss the use case of the trained model. If the model can only be used on the same domain, with object names and numbers identical to the training distribution, the use case seems a bit limited. More experiments on, for example, planning/inference speed compared with different search and heuristics, can be added if the time consumption is a plus. Or any other properties that might come with LLM pre-training, e.g., zero-shot generality.

**Strength And Weaknesses:**

LLMs has shown some impressive results in several language modeling tasks so it would be interesting to see whether they can be useful in planning, and how. This paper is an attempt on the classical planning. The method is quite clear and straightforward, where the pretrained CodeT5 model is fine-tuned on the text sequence which is a compact representation of the domain and problem file. The evaluation on the plan validity and optimality is helpful.
I also have some questions.
- In terms of training and testing data, are the object names randomized in each problem? Is the testing problems having the same object names as the training set? If the initial and goal state of two problem instances has the same set of literals, but in a different order, are they treated as two different problems during training and testing? If swap the object name during testing, will there be a large performance drop?
- In terms of evaluation, are all the baselines tested without fine-tuning? Does the choice of pre-trained model matter a lot to the result? For example, if fine-tuned based on T5, instead of CodeT5, will there be a big performance drop.
- ‘On a relatively more complex domain, i.e., dl, Plansformer-dl attains 76.56% valid plans, out of which 52.61% are optimal. There is a 20% difference between valid and optimal plans for dl showcasing that the model was able to come up with completely new and valid action sequences…’ It’s not clear to me why the difference between valid and optimal plans are new plans. Are the plans in the training set all optimal?
- It seems that, for experiments in 4.3, once fine-tuned on another task, the plan validity on the original problem will drop drastically, many of which drops to 0, according to Fig. 6 in supp materials. Can Plansformer adapt to other domains, while maintaining the ability to solve the problems in the original domain? Or can it generate valid plans if trained on multiple domains simultaneously?
- For these domains, if using lm-cut or h-add heuristics with greedy search, what is the average planning time and percentage of optimal plan?

**Summary Of The Paper:**

This paper showcases the use of LLM on solving classical planning problems. A dataset based on PDDL syntax which includes 4 tasks is generated for training and evaluation of the model. The proposed model, Plansformer, is fine-tuned on the generated dataset from pre-trained CodeT5 model. The model is evaluated on each planning domain using both language modeling metrics and planning validity and optimality metrics. Experimental results show that the model can predict valid plans for the problem and domains it has been trained on.

**Summary Of The Review:**

I think this paper has a straightforward idea and is looking at an interesting problem, but the motivation and use case of the trained model seems unclear to me.  The experiment settings and baselines can be improved to showcase the benefits of learning such a model based off LLMs,  and illustrate the advantages over classical planners.

-----------
Update the score on the revised version.

---

> ### Author Response · Authors · 2022-11-11
> **Addressing feedback from Reviewer abxA**
>
> We thank the reviewer for the comments. Here are our responses to your questions and we hope that we will convince you to see the merits of this paper.
>
> 1. ***In terms of training and testing data, are the object names randomized in each problem? Are the testing problems having the same object names as the training set? If the initial and goal state of two problem instances has the same set of literals, but in a different order, are they treated as two different problems during training and testing? If swap the object name during testing, will there be a large performance drop? =>*** The object names in our test set are not randomized, but we do report Plansformer’s performance on a dataset released in [1], which has different object names as opposed to ours. Plansformer is still able to generate 66% valid plans, whereas their approach achieves 0.6% valid plans on their dataset. This shows that even if we swap the object names, Plansformer will still be able to generate valid plans for the given problem. Same initial and goal state with different ordering are still considered as the same problem, thus eliminating the possibility of duplicates in our dataset.
>
> 2. ***In terms of evaluation, are all the baselines tested without fine-tuning? Does the choice of pre-trained model matter a lot to the result? For example, if fine-tuned based on T5, instead of CodeT5, will there be a big performance drop. =>*** Yes, all the baseline are tested without fine-tuning and the choice of a pre-trained model is important for performance. In our initial testing phase, we have fine-tuned both T5 and CodeT5 with the same blocksworld dataset and hyperparameters and found that fine-tuned T5 gave ~32% valid plans, whereas fine-tuned CodeT5 generated ~90% valid plans. This is because CodeT5 has syntactically meaningful sequences for code-like structured inputs well defined as opposed to T5, which only deals with natural language. With this intuition that models pre-trained on code have an advantage for plan generation, we proceeded with the choice of using CodeT5 for all our experiments.
>
> 3. ***‘On a relatively more complex domain, i.e., dl, Plansformer-dl attains 76.56% valid plans, out of which 52.61% are optimal. There is a 20% difference between valid and optimal plans for dl showcasing that the model was able to come up with completely new and valid action sequences…’ It’s not clear to me why the difference between valid and optimal plans are new plans. Are the plans in the training set all optimal? =>*** Yes, the datasets consist of only optimal plans, thus, the 20% difference between valid plans and optimal plans means that Plansformer was able to come up with new valid action sequences that are not present in its training data.
>
> 4. ***It seems that, for experiments in 4.3, once fine-tuned on another task, the plan validity on the original problem will drop drastically, many of which drops to 0, according to Fig. 6 in supp materials. Can Plansformer adapt to other domains, while maintaining the ability to solve the problems in the original domain? Or can it generate valid plans if trained on multiple domains simultaneously? =>*** Our goal in this paper is to focus on the transfer learning capabilities of language models and check if a model capable of generating valid plans in one domain can transfer its capabilities to solving for plans in a new domain with as limited data as possible. Please see Figure 5 in the main paper and Figures 8-16 in the supplementary.
>
> 5. ***For these domains, if using lm-cut or h-add heuristics with greedy search, what is the average planning time and percentage of optimal plan? =>*** The time reported for FastDownward is using A* LM-Cut heuristic to generate plans. It took 10.28s (see Table 2)
>
> 6. ***The title and some claims in the paper may be revised as the models are mainly tested on one single domain at a time. Specifically, there is no experiment showing one model generating plans on multiple domains. =>*** We have fixed the title of the new paper as per the reviewer’s suggestion to eliminate any ambiguity.
>
> References:
>
> [1] Valmeekam, K., Olmo, A., Sreedharan, S., & Kambhampati, S. (2022). Large Language Models Still Can't Plan (A Benchmark for LLMs on Planning and Reasoning about Change). arXiv preprint arXiv:2206.10498.

---

> > ### Comment · Reviewer_abxA · 2022-11-17
> > **Addressing feedback from author**
> >
> > I appreciate the author's response. I have some follow-up questions/comments.
> >
> > 1. 'Plansformer is still able to generate 66% valid plans, whereas their approach achieves 0.6% valid plans on their dataset.'
> >
> >    Is the 66% result comes from the model that is trained on your dataset and tested on the dataset from [1]? This is not clear in Section 4.2.
> >
> > 2. Fast Downward
> >
> >    It wasn't clear in the original version that A*+LM-cut is used. The revised version has made it clear which is nice. I do think there is an ambiguity when mentioning FastDownward. As in the original paper, it's using the causal graph heuristics and some kind of best-first search, which I think do not produce an optimal plan. I think it'll be helpful to clarify that up a bit.
> >
> > 3. 'Our goal in this paper is to focus on the transfer learning capabilities of language models and check if a model capable of generating valid plans in one domain can transfer its capabilities to solving for plans in a new domain with as limited data as possible.'
> >
> >     If the major point is that a model trained on one domain, can learn to generate plans in another domain with fewer data, then a baseline with a model trained on another domain, with the same amount of truncated data (e.g. 500, 1000) should be added. Currently in Figure 5, there is a comparison between models that are pre-trained on another domain, and the one that is not, with all the data. For results like bw-hn[500], there is no direct comparison that can show the benefit of transfer learning.
> >
> >     The conclusion 'Failed plans keep decreasing with additional fine-tuning data points for all domains.' is fairly intuitive. My concern is that, if finetuning on another domain leads to a 0% valid plan on the original domain (as shown in the result in the supp material), the generalization capability can be something to worry about.

---

> > > ### Author Response · Authors · 2022-11-18
> > > **Addressing additional feedback from Reviewer abxA**
> > >
> > > Thank you for going through our rebuttal and asking clarifying questions. We wish to address the additional feedback.
> > >
> > > 1. ***'Plansformer is still able to generate 66% valid plans, whereas their approach achieves 0.6% valid plans on their dataset.'
> > > Is the 66% result comes from the model that is trained on your dataset and tested on the dataset from [1]? This is not clear in Section 4.2.*** => 66% comes from Plansformer trained on our dataset and tested on the dataset given in [1]. This will be added in Section 4.2 to remove ambiguity.
> > >
> > > 2. ***I do think there is an ambiguity when mentioning FastDownward. As in the original paper, it's using the causal graph heuristics and some kind of best-first search, which I think do not produce an optimal plan. I think it'll be helpful to clarify that up a bit.*** => For STRIPS-style planning tasks, which include all the 4 domains under consideration, A* + LM-Cut provides optimal plans as observed in [2]. This reference will be added in Section 4.2 of the revised paper.
> > >
> > > 3. ***On additional experiments*** => We ran additional experiments to develop a single Plansformer model trained on all four domains and tested on these four domains. The test bed on which this single Plansformer model is tested is the same as the test bed used by Plansformer-bw, Plansformer-hn, Plansformer-gr, and Plansformer-dl, i.e., each domain's test bed consists of 3000 problem instances. The results obtained are as follows -
> > >
> > > | Test Domain | Valid Plans | Cost-Optimal Plans | Invalid Plans                       |
> > > |-------------|-------------|--------------------|-------------------------------------|
> > > | **bw**      | 95.83%      | 93.75%             | Failed = 4.17%, Incomplete = 0%     |
> > > | **hn**      | 79.25%      | 76.72%             | Failed = 2.34%, Incomplete = 18.41% |
> > > | **gr**      | 78.44%      | 50.61%             | Failed = 15.03%, Incomplete = 6.53% |
> > > | **dl**      | 63.03%      | 55.25%             | Failed = 2.81%, Incomplete = 34.16% |
> > >
> > > The single Plansformer trained on all models outperforms Plansformer-bw by 5.79% (in terms of valid plans) and has around a 4.5% to 13.5% decrease in valid plans for the other three domains. We are also performing the additional experiments requested to show transfer learning effectively. We will update those results soon and will add all these details in the final version of the paper.

---

> > > > ### Comment · Reviewer_abxA · 2022-11-18
> > > > **Addressing feedback from author**
> > > >
> > > > Thanks for the clarification and the additional experiment. This is very helpful.
> > > >
> > > > Regarding the planning and heuristics, I think the ambiguity is not on whether A*+LM-Cut produces the optimal plan, but on whether the causal graph heuristics+best first search as in the original FastDownward paper is used. I think the answer is no. It may be helpful to clarify that a bit in the writing.

---

### Official Review · Reviewer_X39q · 2022-10-24

**Confidence:** 4
**Correctness:** 4
**Technical Novelty And Significance:** 3
**Empirical Novelty And Significance:** 3
**Recommendation:** 8

**Clarity, Quality, Novelty And Reproducibility:**

The line of work is a popular line of investigation. Focusing on symbolic planning is novel and almost a necessary question.
The manuscript is clear enough for writing a review but it should be revised. For instance,
- there is some confusion on phases and stages.
- "All Plansformer models generate 0% incomplete generations for respective domains". Is that rounding down 0.x%?

The core of the evaluation is clear, and the observations are not trivial.

**Strength And Weaknesses:**

Strengths
- Symbolic planning seems to offer a challenging problem for code generation in new domains. Even if it can be argued that such code is different from the usual one, the sequence of actions could as well correspond to actions in a robot or an API call
- Evaluating using ROUGE and BLEU allows seeing the behaviour of LLM with a popular metric.
- The planning problems are reasonably simple

Weaknesses
- One can argue that zero-shot is not the recommended methodology for testing an LLM in a domain like this. Many reasoning tasks are evaluated including one demonstration.
	- I understand it's cumbersome to find the best prompt for them. However, I suspect the readers would prefer some prompt investigation. The fine-tuning time is short (at most 75mins), so the computing cost might not become too high.
- In contrast, each model is specialized in a domain.
	- Moreover, the transfer to a new domain required a significant amount of examples to start observing significant performance. Providing one example seems only fair.
- The size of the models is not said explicitly.
	- LLMs has shown additional generalization at larger sizes. Perhaps the conclusions would change with bigger models. I
- Single set hyper-parameters (Appendix sect 3.2).
	- One can argue the hyper-parameters are to be fixed across experiments, but the models are being tuned.
	- See question below on hyper-parameters

Non-Weaknesses (these are potential weak points that I think don't diminish the submission)
- Code evaluation is usually complemented by constrained decoding.
	- When grammar is available, LLMs tend to be used with constrained decoding. I understand this is not the point of the paper as none of the methods used that, but it should be discussed in related work and make an explicit statement.

Additional comments
- I miss an explicit token to indicate the beginning of the task.
	- For instance, in Fig 4, CodeT5 might be predicting the parameters of part of an effect in the prompt.
	- I wonder what would happen with an explicit token plus a demonstration.
- Plan lengths are not reported. Instances with longer plans might be more prone to testing.
- I suspect the 80-20% splits might be helping the proposed model.
	- The problems for testing are in the same distribution, although they are rather small.
	- The 20% might have share sub plans with the training data.
	- If that's the case, perhaps the proposed method would not actually help to plan in a new domain. I'm ok with that, but the conclusions of the paper should be toned down.
- Please clarify in the main body
	- Plansformer (no suffix) in Table 1 is 4 different models. I suggest adding an explicit column on the data used for evaluation.

Questions:
- What pre-trained models were used precisely?
	- How many parameters?
	- What hyper-parameters were used for testing them?
- How similar are the sequences of the training vs the testing data?
	- Perhaps a pair-wise comparison using distance can reveal some patterns.
	- Is the generation repeating objects in the same order? There might exist artifacts that wouldn't affect a classical planner but would affect LLMs.
- Is CodeT5 slower just because it is generating a longer sequence?
	- We are missing hyper-parameters for them.


**Summary Of The Paper:**

The paper study considers the question of whether zero-shot large pre-trained language models (LLMs) can generate plans by studying performance in symbolic planning problem domains. For each sample, the model receives the goal, the initial state and the available actions. The model used for fine-tuning is popular for code generation. The empirical results show that although some models are frequently able to generate sequences similar to the gold truth, they are not valid plans.

In the main body, I concluded –but it wasn't clear– that no model was trained using the data from all the domains. Fig 4 in the appendix confirms that.


---

increasing scores after the updates

**Summary Of The Review:**

The paper study how a LLM specialized in code generation cannot generate symbolic plans without specifying tunning. The contribution is two-fold: on one hand, the paper identifies symbolic planning as an interesting benchmark. Second, it proposes a method for achieving high-performing models, although they require a symbolic planner. This motivates researchers on LLMs for reasoning to attempt to get similar performance using other tools.

I'm now more inclined to accept the paper, but it depends on the answers to the questions.

---

> ### Author Response · Authors · 2022-11-11
> **Addressing feedback from Reviewer X39q**
>
> We thank the reviewer for the feedback and for the time spent on our manuscript. We reply to the reviewer’s concerns below.
>
> 1. ***On prompt investigation for plan generation =>*** [1] explores the use of prompting for generating plans and shows that it is not effective. On their dataset, prompting GPT-3 obtains 0.6% valid plans, whereas Plansformer generates 66% valid plans (refer to Section 4.2).
>
> 2. ***LLMs have shown additional generalization at larger sizes. Perhaps the conclusions would change with bigger models. =>*** We do have one of the largest code generation models with 12 billion parameters, Codex, in our comparison tables (refer to Table 1 and Table 2). Even with 12B parameters in Codex. Plansformer obtained 90% valid plans compared to 0.15% by Codex.
>
> 3. ***When grammar is available, LLMs tend to be used with constrained decoding. I understand this is not the point of the paper as none of the methods used, but it should be discussed in related work and make an explicit statement. =>*** We briefly mentioned this point in the related work (Section 2.2).
>
> 4. ***Plan lengths are not reported. =>*** The average plan length for blocksworld is 9, driverlog is 10, gripper is 9, and hanoi is 12. This information has been added in Section 3.1.1 of the revised paper.
>
> 5. ***I suspect the 80-20% splits might be helping the proposed model =>*** We have adopted a 5-fold validation approach for computing the results to make sure we are not biasing the test set to have sub-plans appearing in the training set. These details have been added in Section 3.1.1 and Section 4 of the revised paper for further clarification.
>
> 6. ***Plansformer (no suffix) in Table 1 is 4 different models. I suggest adding an explicit column on the data used for evaluation. =>*** Plansformer without a suffix in Table 1 is the mean of Plansformer-bw (tested on blocksword), Plansformer-hn (tested on hanoi), Plansformer-gr (tested on gripper), and Plansformer-dl (tested on driverlog). This information has been mentioned in Section 4.1 of the paper.
>
> 7. ***On parameters and hyperparameters for pre-trained models =>*** Codex has 12 billion parameters, GPT-2 has 1.2 billion parameters, T5-base has 220 million parameters, and CodeT5-base has 8.35 million parameters. The hyperparameters for all the models for plan generation are the same as that of Plansformer mentioned in Section 3.2 of supplementary material.
>
> 8. ***"All Plansformer models generate 0% incomplete generations for respective domains". Is that rounding down 0.x%? =>*** Yes, we rounded the percentages to ~0%.
>
> References:
>
> [1] Valmeekam, K., Olmo, A., Sreedharan, S., & Kambhampati, S. (2022). Large Language Models Still Can't Plan (A Benchmark for LLMs on Planning and Reasoning about Change). arXiv preprint arXiv:2206.10498.

---

> > ### Comment · Reviewer_X39q · 2022-11-22
> > **Thank you**
> >
> > Thank you for your comments.
> >
> > I think it'd be good to add prompts to this experimental setting, as this is different from [1].

---

### Official Review · Reviewer_T1dJ · 2022-10-26

**Confidence:** 4
**Correctness:** 2
**Technical Novelty And Significance:** 3
**Empirical Novelty And Significance:** 2
**Recommendation:** 3

**Clarity, Quality, Novelty And Reproducibility:**

**Clarity**

Some parts of the paper are clear enough, but most of the paper is very poorly written. I hope these problems are corrected in the rebuttal version.

Why is Rogers et al. referenced on the first line of the intro?

Most references are improperly parenthesized.

Poorly worded sentences like the following make the reader’s job much harder than it needs to be:
- “promisingly tell us the failures”
- “with a few samples than learning from scratch”
- “due to the fine-tuneup of the models”
- “We call the model with new weights that can generate symbolic plans as Plansformer.”
- “The two stages mentioned in the evaluation phase checks for finding out the efficiency of Plansformer as a language model and a planner.”
- “Plansformer… requires to be tested for plan validation”
- “A drop in the number of optimal plans reduces with the increasing complexity of the domains.”
- “200x faster than the ground truth”  Instead, this should say “200x faster than the FastDownward planner, which generated the ground truth plans”

**Quality**

The overall quality of the work is quite good.

**Novelty**

Fine-tuning of LLMs for planning appears to be novel and timely.

**Reproducibility**

The code and fine-tuned models would be very useful, but I didn’t see any mention that they would be provided.


**Strength And Weaknesses:**

**Strengths**

The work is well-executed, and the results are noteworthy. On four different planning domains, Plansformer significantly outperforms baseline LMs, and sometimes reaches 90% plan validity without the high computational cost of the baseline planner (FastDownward).

The adaptation experiments give interesting results, showing that a second round of fine-tuning (on a domain different than the first) produces a model which is at least as accurate as training on the second domain alone. This may indicate some degree of successful transfer between domains. But the missing experiment would be to fine-tune a single model for twice as long on the same domain, which would presumably produce the best results of all for each domain.

The “model-testing” evaluations (in Table 1) are useful as sanity checks, but they are not nearly as important as the validity and optimality tests, and could be moved to an appendix if space runs short.

**Weaknesses**

The paper says “There isn’t much prior work to adopt as a baseline and perform a comparison with Plansformer.” But in fact, deep neural networks and even transformers have been applied to planning problems, including those described by PDDL:
- DeConNet: Deep Neural Network Model to Solve the Multi-Job Assignment Problem in the Multi-Agent System, Lee et al., 2022, https://www.mdpi.com/2076-3417/12/11/5454
- Learning to Delegate for Large-scale Vehicle Routing, Li et al., 2021, https://arxiv.org/abs/2107.04139
- PDDLGym: Gym Environments from PDDL Problems, Silver and Chitnis, https://arxiv.org/abs/2002.06432

Including non-LLM baseline models such as these in the experiments would provide a more accurate gauge of Plansformer’s abilities.

The paper says: “Plansformer uses masked language modeling, attending to tokens on either side of the masked word.” This gives the impression that Plansformer is just a transformer encoder, when in fact it is an encoder-decoder pair, as is CodeT5. And Plansformer’s decoder uses standard causal attention instead of bidirectional masked attention. Furthermore, the paper gives no details of bidirectional masked pretraining at all. Was it performed? Or was all of this work’s fine-tuning done on the decoder’s autoregressive generation of plans?

It is probably the case that each problem description is passed to Plansformer’s encoder, then the plan is produced by the decoder. But this basic flow of operation is never explicitly stated in the paper.

Because Plansformer’s plans always fall well short of 100% validity and optimality, the bolded phrase in the following sentence appears to be incorrect:  “We then compared its behavior to an existing state-of-the-art planner, showing that our LLM-based planner, called Plansformer, **can solve many more instances with high quality both in terms of correctness and length** while needing much less time to generate such plans.”

“88.44% are optimal and have the highest performance compared to **other domains**.” But as explained in other parts of the paper, optimal means that the plan is the shortest possible (minimum cost). What does this have to do with “other domains”?

“All Plansformer models generate 0% incomplete generations for respective domains.” This seems to contradict Fig. 3, which shows an incomplete plan generated by Plansformer on Blocksworld.

Fig. 5 and others like it would be less confusing if the first bar were removed, because it doesn’t represent testing on the same domain as all the other bars in the chart.


**Summary Of The Paper:**

Continuing the successful pattern of fine-tuning large LMs on specialized domains, this paper presents Plansformer, a LLM (CodeT5) which is additionally fine-tuned to generate plans that solve problems expressed in PDDL. Plansformer is found to generate plans with far higher validity than LLMs (including CodeT5 itself) without fine-tuning on these domains. And Plansformer generates plans with far greater speed than exhaustive planners like FastDownward (though at a reduced validity rate).

**Summary Of The Review:**

The Plansformer is a promising application of large language models to planning problems. But this message may be lost on the community if the clarity of the paper’s writing is not significantly improved.

***** POST-DISCUSSION UPDATE*****

The authors are to be commended for performing additional experiments on the impact of object name randomization, and providing a table of results from randomizing the object names for either train or test or both. In my view, this new table and the paper's Table 2 tell a complex story. When train and test use the **same** naming scheme (either original or fully randomized), then the impact of object names on valid plans is highly task-specific, ranging from a very small drop (0.21%) on Blocks World, to a significant drop (15.41%) on Hanoi. But when train and test **do not** employ the same object naming scheme, validation performance falls dramatically, from 76.56% to 2.69% in the worst case.

My conclusion is that the issue of object name randomization cannot be safely ignored, because it calls into serious question the proper interpretation of the previous results (without object name randomization). The only solution I see is to repeat all of the experiments using (2-character) object name randomization. My expectation is that the paper’s primary conclusions will still hold up, but I don’t think the experiments can be skipped.

For these reasons, I regret having to leave my score unchanged.

---

> ### Author Response · Authors · 2022-11-11
> **Addressing feedback from Reviewer T1dJ**
>
> We thank the reviewer for the helpful comments. We address your questions below:
>
> 1. ***The paper says “There isn’t much prior work to adopt as a baseline and perform a comparison with Plansformer.” But in fact, deep neural networks and even transformers have been applied to planning problems, including those described by PDDL:
> (a) DeConNet: Deep Neural Network Model to Solve the Multi-Job Assignment Problem in the Multi-Agent System, Lee et al., 2022, https://www.mdpi.com/2076-3417/12/11/5454
> (b) Learning to Delegate for Large-scale Vehicle Routing, Li et al., 2021, https://arxiv.org/abs/2107.04139
> (c) PDDLGym: Gym Environments from PDDL Problems, Silver and Chitnis, https://arxiv.org/abs/2002.06432
> Including non-LLM baseline models such as these in the experiments would provide a more accurate gauge of Plansformer’s abilities. =>*** Our work was focussed on demonstrating if a transformer architecture is capable of generating valid plans for multiple domains. Among the suggested works, Paper (a) is the closest but the implementation is limited to working on a multi-agent transportation domain, which would limit our experimentation in testing the model for other domains in consideration. Paper (b), however, solves a different task of subgoal identification and not plan generation. Paper (c), has different PDDL domains with only a few problem instances per domain represented in sequential decision making format, and does not deal with plan generation.
>
> 2. ***On clarification regarding the use of masked language modeling for Plansformer and bidirectional masked pretraining =>*** This sentence present in Section 4.1 has been revised in the latest version of the paper to remove ambiguity. Also, we make use of decoder’s autoregressive nature to generate plans and do not perform bidirectional masked pretraining.
>
> 3. ***Because Plansformer’s plans always fall well short of 100% validity and optimality, the bolded phrase in the following sentence appears to be incorrect: “We then compared its behavior to an existing state-of-the-art planner, showing that our LLM-based planner, called Plansformer, can solve many more instances with high quality both in terms of correctness and length while needing much less time to generate such plans.” =>*** This sentence has been corrected in Section 5 of the revised version of the paper.
>
> 4. ***“88.44% are optimal and have the highest performance compared to other domains.” But as explained in other parts of the paper, optimal means that the plan is the shortest possible (minimum cost). What does this have to do with “other domains”? =>*** Here, it is a mere comparison with other Plansformer variants for respective domains and to point out which Plansformer variant has the highest performance in terms of optimality. In the latest revision, we have removed the phrase “compared to other domains” to eliminate any ambiguity.
>
> 5. ***“All Plansformer models generate 0% incomplete generations for respective domains.” This seems to contradict Fig. 3, which shows an incomplete plan generated by Plansformer on Blocksworld. =>*** We stand corrected in this claim. It should have been “All Plansformer models generate ~0% failed plans for respective domains.”. We have also defined what an incomplete/wrong generation and a failed plan is, in addition to correcting this sentence in the latest revision of the paper (Section 4.2).
>
> 6. ***Fig. 5 and others like it would be less confusing if the first bar were removed, because it doesn’t represent testing on the same domain as all the other bars in the chart. =>*** Figure 5 and others (belonging to supplementary material) have been corrected to accommodate this suggestion.
>
> 7. ***On overall clarity of the paper =>*** All the mentioned suggestions that would lead to better clarity in the paper have been accounted for in the revised version of the paper.
>
> 8. ***On release of code and fine-tuned models =>*** The dataset, code, and all fine-tuned model checkpoints will be released for public usage. This is mentioned as a footnote on Page 1 in the revised version of the paper.

---

> > ### Comment · Reviewer_T1dJ · 2022-11-17
> > **Addressing the authors' response**
> >
> > The authors did respond to most of my concerns, and made improvements to most of the poorly worded sentences that I pointed out. But the resubmitted paper’s writing quality is still so poor overall that I feel it will require substantial improvements in presentation before most readers can benefit from it. The paper requires *much* more than another proof-read. And most citations are still parenthesized incorrectly, contrary to the response (to Reviewer CWqr) that they were fixed.
> >
> > In response to the question of Reviewer abxA regarding the randomization of object names between train and test, the authors responded that they are *not* randomized. This surprised me, since it largely invalidates the paper’s experimental results. The difference between Plansformer’s performance of 66% vs the 60% of the approach of [1] is not significant enough to make up for this methodological error.
> >
> > In view of these problems, I’m lowering my score from 6 to 3.

---

> > > ### Author Response · Authors · 2022-11-17
> > > **Addressing feedback from Reviewer T1dJ**
> > >
> > > We thank Reviewer T1dJ for going through our rebuttal comments. We want to clarify about randomization of object names between train and test sets. We are considering two kinds of randomization:
> > > 1. Changing the ordering of the predicates along with object names. For example, {on-table b1, on-table b2, clear b3} and {clear b3, on-table b1, on-table b2} are two pairs with a different ordering.
> > > 2. Changing the object names itself that are associated with the predicates. For example, {on-table b1} and {on-table a}. Here, b1 and a reference to the same block.
> > >
> > > Plansformer works with both of the above two different kinds of randomization. We observed that changing the ordering of the predicates along with object names does not affect Plansformer's ability to generate valid plans. Plansformer can also generate valid plans when the object names are changed. To support this, we report Plansformer's performance on the dataset present in [1], which has different object names as opposed to the ones used in our dataset. There is a misunderstanding of the performance reported for our approach compared to the approach in [1]. Plansformer generates 66% valid plans on the dataset released in [1], whereas their approach produces 0.6% (less than 1%) valid plans, thereby showing a significant improvement in our approach.
> > >
> > > Regarding the citations being incorrectly parenthesized, we have used the ICLR-released template to add citations to our paper. We want to correct if we have made any mistakes, and it would help us if an example could be given of how the citations should appear.

---

> > > > ### Comment · Reviewer_T1dJ · 2022-11-18
> > > > **Thank you**
> > > >
> > > > Thank you for the clarification regarding 60% vs. 0.6%.
> > > >
> > > > Regarding citations, the problem is not with the ICLR template, it's that your paper seems to use \citet{} for all citations, even for cases where \citep{} is appropriate. For example, the first few citations in paragraph 2 of the Introduction should use \citep{}. Please consult the web to understand the difference.

---

> > > > > ### Author Response · Authors · 2022-11-20
> > > > > **Addressing prior comments**
> > > > >
> > > > > In our initial response, we addressed some of the questions and would like to respond to the reviewer’s other questions in this reply. We have made revisions to the paper to improve its quality to the best of our ability. We have also fixed all the parenthesization issues in the citations as advised by the reviewer. We hope the reviewer finds the revised paper satisfactory with the feedback/questions accounted for.
> > > > >
> > > > > 1. ***The flow of operation between encoder and decoder =>*** As pointed out, the encoder generates features for the given problem instance as input. The generated features from the encoder are passed on to the decoder to produce the plan as output. This has been added in Section 3.1.3 of the revised paper.
> > > > >
> > > > > 2. ***But the missing experiment would be to fine-tune a single model for twice as long on the same domain, which would presumably produce the best results of all for each domain. =>*** Increasing the number of data points and training the model for twice as long would probably produce the best results. Nevertheless, the computational cost and time to produce more data points for complex domains such as driverlog and gripper are very high. Thus, we wanted to see if a language model can generate valid plans using as little data as possible. However, to accommodate Reviewer abXa’s feedback, we trained a single model on all four domains under consideration. The results are as follows -
> > > > >
> > > > > | Test Domain | Valid Plans | Cost-Optimal Plans | Invalid Plans                       |
> > > > > |-------------|-------------|--------------------|-------------------------------------|
> > > > > | **bw**      | 95.83%      | 93.75%             | Failed = 4.17%, Incomplete = 0%     |
> > > > > | **hn**      | 79.25%      | 76.72%             | Failed = 2.34%, Incomplete = 18.41% |
> > > > > | **gr**      | 78.44%      | 50.61%             | Failed = 15.03%, Incomplete = 6.53% |
> > > > > | **dl**      | 63.03%      | 55.25%             | Failed = 2.81%, Incomplete = 34.16% |
> > > > >
> > > > > The single Plansformer trained on all models outperforms Plansformer-bw by 5.79% (in terms of valid plans) and has around a 4.5% to 13.5% decrease in valid plans for the other three domains. These details have been added in Section 4.2.2 of the supplementary material.

---

> > > > > > ### Comment · Reviewer_T1dJ · 2022-11-21
> > > > > > **Thank you**
> > > > > >
> > > > > > I appreciate the extra work and valuable information that the authors provided today. I’ve read all their responses, as well as the final version of the paper and appendix. It is certainly much improved from the initial submission, and the new experiments are very interesting.
> > > > > >
> > > > > > I see the lack of object name randomization as a fundamental problem. Since object names (like b2 or a) are arbitrary, and carry no intrinsic importance, the standard approach (when using limited datasets like these) is to randomize the names for every example in both the train and test sets. Failure to randomize in this way is a methodological error, especially when transformers are involved, given their proven ability to cheat by memorizing unimportant details. (By contrast, name randomization can usually be safely skipped when using web-scale data.) It’s obviously too late to repeat all of the experiments reported in this work, so I commend the authors for doing the next best thing, which was to perform a smaller-scale test on out-of-domain object names. There, Plansformer’s performance of 66% vs. the 0.6% of [1] suggests that Plansformer’s performance might not have degraded too badly on this paper's main experiments if the object names had been properly randomized throughout. But regrettably, despite this positive sign, the absence of randomization still prevents us from drawing solid conclusions regarding this work’s main results. I look forward to seeing Plansformer results that are not clouded by this issue.
> > > > > >
> > > > > > While the authors have labored to fix the paper’s wording and have clarified many issues, I believe that coming to understand the paper is still too much work for the reader, who has to figure out too many puzzles along the way. Regarding citations, where only \citet{} was used before, now \citep{} appears to be used everywhere. But some of the citations need to use \citet{}, such as all works cited in the final paragraph of section 2.2. Or else the surrounding wording needs to change to make \citep{} appropriate. These are all things that more time and better technical editing can correct.
> > > > > >
> > > > > > Since I’m excited by so many aspects of this work, and it represents such impressive efforts, I regret having to leave my score at 3.

---

> > > > > > > ### Author Response · Authors · 2022-11-29
> > > > > > > **Notifying regarding updated work**
> > > > > > >
> > > > > > > Dear Reviewer,
> > > > > > >
> > > > > > > Thanks for the previous feedback. We have posted insights from additional randomization experiments and addressed other outstanding issues in our latest general response. Please let us know if we can answer any other clarification questions to improve the paper and help with the decision.

---

### Official Review · Reviewer_CWqr · 2022-10-26

**Confidence:** 4
**Clarity, Quality, Novelty And Reproducibility:** 1. There are many typos and incorrect…
**Correctness:** 3
**Technical Novelty And Significance:** 3
**Empirical Novelty And Significance:** 3
**Recommendation:** 5

**Strength And Weaknesses:**

Strengths

1. The motivation of this paper is well-formed and the experimental setup is quite sound—the authors create a dataset to fine-tune a language model on, and then evaluate how well the model can reliably produce plans that are syntactically and semantically correct and executable.
2. They evaluate using standard NLP metrics like ROUGE/BLEU—although it might be worth looking at more executable metrics like measure syntax (e.g., through the underlying grammar) or semantic aspects (e.g., through the propositions or predicates being referenced) rather than string-match metrics.
3. The conclusion and ongoing work sections have interesting future work directions that draw off of this paper—however, some clarity on the future work could be helpful. Can the authors elaborate on the “learning planner” idea maybe in it’s own section, or otherwise remove from the conclusion section?

Weaknesses / Questions

1. Can a navigation-like domain also be handled by this model? It might be worth adding a domain that is different from the existing ones (in terms of its action space) to illustrate the generality of this training mechanism to show that the model can generate plans in that domain as well.
2. What do the train/test splits look like in terms of distributional differences? From reading the paper, it sounds like the splits were made randomly? My main concern (especially from looking at the results which look very high) is that the test splits are almost identical, or at least not very structurally different, from the train splits, which means the model does not actually have to generalise very much to perform well on the test splits.
3. Along those lines, can we re-create the test split to test the following. (a) if the test plans are of length >>> the length of plans in the train dataset (b) if the nested strcture/syntax of test plans is deeper/more complex than those in the train set (c ) if some set of objects/predicates are held out in samples in the test set
4. I would expect performance to at least decrease a little on these splits—and this is what we want to test to truly measure test-time generalisation and whether the model really is learning the syntax/semantics of plans. But if the performance is significantly lower, this is an important thing to evaluate and put into the paper, so I would like to see this table of results.
5. There are lots of things that warrant some more explanation/clarity: what is FastDownward and is it crucial to the process? If it is, there should be a small paragraph that briefly describes it as well as it’s relevance to the generated dataset.
6. It is also unclear what the dataset looks like for the different domains, so Figure 2 could be updated to contain smaller snapshots for different domains to make it clearer.
7. Can we have more details of the fine-tuning process even if only in the supplementary material, because there currently does not exist information about fine-tuning and what layers/components of the model are being updated.
8. When talking about evaluation in terms of time taken to solve the test-bed, can you clarify why the models take longer? Is it an inference-time generation problem or are there other reasons that it might take longer?



**Summary Of The Paper:**

This paper attempts to use large language models for automated planning, by fine-tuning them on samples of plans (sequences of actions / a path determined by a planner). They essentially frame plan-formation as a supervised learning task, relying on the capability of a pre-trained language model to learn and adapt to the planning language’s syntax in the domains they test in. They create such a dataset to fine-tune the models on and they show that this fine-tuned language model (that they call “plansformer”) is capable of generating plans for 4 different domains or tasks they test on.


**Summary Of The Review:**

This paper is well-motivated and executed, however the evaluation (mainly the test datasets) might not be as sound/correctly generated as needed to properly evaluate generalisation. If the authors can redo this and add a more concrete evaluation this would be a very nice paper with lots of interesting insights!

---

> ### Author Response · Authors · 2022-11-11
> **Addressing feedback from Reviewer CWqr**
>
> We thank the reviewer for the insightful comments and suggestions. Here we answer some of the questions raised by the reviewer.
>
> 1. ***On elaborating “learning planner” in the Conclusion and Ongoing Work section =>*** We have elaborated on the term “learning planner” in Section 5 in the revised version of the paper.
>
> 2. ***On whether a navigation-like domain can be handled by Plansformer =>*** driverlog is a navigation-like domain, involving moving packages between locations on trucks driven by drivers. This domain is quite different from the other three domains considering action space.
>
> 3. ***On distribution difference in train-test split =>*** The training dataset itself consists of varying objects/configurations for each considered domain. For example, the blocksworld dataset has 2 to 5 block configurations. The train-test split is done using 5-fold cross-validation to mitigate the possibility of bias. This information has been added to the paper in Section 3.1.1 and Section 4. Additionally, we have also tested our Plansformer model on a different blocksworld dataset that was released in [1] and obtained 65.4% more valid plans than them.
>
> 4. ***On generating train-test split with different plan lengths of test set greater than train set =>*** In order to answer this, we have performed additional experiments. We have created a train set for blocksworld consisting of 2,3 block configurations and a test set with 4,5 block configurations. The test set consists of 100 total instances, 50 from 4 block configuration and 50 from 5 block configuration. The average plan length for the train set and test set is 4 and 10 respectively. Plansformer trained on 2,3 block configurations was able to generate 64% valid plans on problem instances with 4,5 block configurations from the test set, showing that our approach can generate valid plans even if the plan length for test set >>> train set. We were able to achieve 64% valid plans in this experimentation as the train set consists of only 162 data points (all possible 2,3 block configurations) as compared to 87% obtained by Plansformer-bw (trained on 14,400 data points) on the same test set. This information has been added in Section 4.2 of the supplementary material.
>
> 5. ***On clarity regarding FastDownward and its relevance to the dataset  =>*** FastDownward is a classical planning system based on heuristic search, searching the space of world states of a planning task in the forward direction. It won the "classical (i.e., propositional, non-optimising) track” of the 4th International Planning Competition at ICAPS 2004. It is capable of generating optimal plans when used with A* LM-Cut heuristic, thereby, prompting us to use it for generating a planning dataset consisting of optimal plans. The revised version of the paper is updated with this information in Section 3.1.1.
>
> 6. ***On having snapshots of datasets for other domains apart from blocksworld =>*** Figure 1 in supplementary material shows snapshots for all the considered domains in the paper.
>
> 7. ***Information about the architecture of Plansformer and what layers are being updated during fine-tuning =>*** Section 5 in supplementary material of the revised paper has details regarding the architecture (layers) of Plansformer.
>
> 8. ***Long time taken for plan generation by other models  =>*** The reason for this is mentioned in Section 4.2 with an example showcased in Figure 4. The reason for longer time is due to the models not having sufficient knowledge about what a plan is, including its validity and optimality.
>
> 9. ***On incorrect citations and typos =>*** All the mentioned citations and typos have been fixed in the revised paper. We have also proof-read the paper and made sure all errors are addressed.
>
> References:
>
> [1] Valmeekam, K., Olmo, A., Sreedharan, S., & Kambhampati, S. (2022). Large Language Models Still Can't Plan (A Benchmark for LLMs on Planning and Reasoning about Change). arXiv preprint arXiv:2206.10498.

---

> > ### Author Response · Authors · 2022-11-28
> > **Requesting for feedback or further clarification questions**
> >
> > We hope we have addressed all the questions and other outstanding issues. Are there other clarification questions we can answer to improve the paper and help with your decision?

---

### Author Response · Authors · 2022-11-20
**Common Response to Reviewers**

We thank all reviewers for the helpful feedback and comments! We describe the common concerns here and in addition,  have addressed each comment individually.

In terms of written improvement, we have **(1)** revised the paper for readability, **(2)** changed the paper's title to "Plansformer: Generating Symbolic Plans using Transformers.", **(3)** added additional experimentation details in Section 4 and also added sections 4.2.1, 4.2.2, and 5 in the supplementary material, and **(4)** fixed the parenthesization issues in the citations.

Additionally, we have performed three sets of experiments and shown significant insights. They are, **(1)** Train-test split, where we describe how Plansformer can perform on a test set with plan lengths longer as opposed to the train set and also handle object names randomization *(Section 4.2 of supplementary material)*, **(2)** Transfer learning, where we had shown that a Plansformer base model, when further fine-tuned on another domain, has a significant increase in performance in comparison with a Plansformer base model belonging to the testing domain at all the considered data points *(48% difference between Plansformer-bw-dl[5000] and Plansformer-dl[5000]; see Section 4.2.1 of supplementary material)*, and **(3)** A single model for all domains (multi-domain plan generation), where we found that the single Plansformer model has relatively comparable performance to the base models *(4.5% to 13.5% decrease in valid plans; see Section 4.2.2 of supplementary material)*.

We now describe the details of the experiments below.

1. ***A number of reviewers had questions regarding the train-test split of the dataset, randomization of object names and generating train-test split with different plan lengths of test set greater than train set =>*** The training dataset consists of varying objects/configurations for each considered domain to have unique problem instances with different complexities. The train-test split is done using 5-fold cross-validation to mitigate the possibility of bias. This information has been added to the paper in Section 3.1.1 and Section 4.
When it comes to the randomization of object names, we are considering two different possibilities:
    + Changing the ordering of the predicates along with object names. For example, {on-table b1, on-table b2, clear b3} and {clear b3, on-table b1, on-table b2} are two pairs with a different ordering.
    + Changing the object names itself that are associated with the predicates. For example, {on-table b1} and {on-table a}. Here, b1 and a reference to the same block.

Plansformer works with both of the above two different kinds of randomization. We observed that changing the ordering of the predicates along with object names does not affect Plansformer's ability to generate valid plans. Plansformer can also generate valid plans when the object names are changed. To support this, we report Plansformer's performance on the dataset present in [1], which has different object names than those used in our dataset. Plansformer trained on our dataset and tested on the dataset in [1] generated 66% valid plans, whereas their approach produces 0.6% (less than 1%) valid plans, thereby showing a significant improvement in our approach.

In order to test Plansformer’s ability to generate valid plans if the train set has smaller plans as opposed to the test set, we created a new train set for blocksworld consisting of 2,3 block configurations and a test set with 4,5 block configurations. The test set consists of 100 instances, 50 from 4 block configuration and 50 from 5 block configuration. The average plan length for the train and test sets is 4 and 10, respectively. Plansformer trained on 2,3 block configurations was able to generate 64% valid plans on problem instances with 4,5 block configurations from the test set, showing that our approach can generate valid plans even if the plan length for test set >>> train set. We achieved 64% valid plans in this experimentation as the train set consists of only 162 data points (all possible 2,3 block configurations) compared to 87% obtained by Plansformer-bw (trained on 14,400 data points) on the same test set. This information has been added in Section 4.2 of the supplementary material.

---

> ### Author Response · Authors · 2022-11-20
> **Common Response to Reviewers (contd.)**
>
> 2. ***Few reviewers had clarifying questions regarding FastDownward planner and its significance in building the dataset =>*** FastDownward is a classical planning system based on heuristic search, searching the space of world states of a planning task in the forward direction. It won the "classical (i.e., propositional, non-optimizing) track" of the 4th International Planning Competition at ICAPS 2004. There are multiple search algorithms available at the disposal of FastDownward by default. We make use of the A* LM-Cut heuristic as it is capable of generating optimal plans [2]. Thus, we use FastDownward to generate a planning dataset consisting of optimal plans. The revised version of the paper is updated with this information in Section 3.1.1.
>
> 3. ***Few reviewers requested additional experiments to provide more evidence for the advantage of transfer learning and also wanted to know the performance of a single model trained on all four planning domains =>*** Initially, in Figure 5 of Section 4.3, we only showed the graphs by fine-tuning a base model (Plansformer-bw) with varying data points of the other three domains to check for transfer learning and compared the performance with the corresponding domain's base model (i.e., a model trained on 14,400 training data points in that particular domain). However, as per the received feedback, we further trained base models with varying data points similar to the fine-tuning approach mentioned above for the domains on which transfer learning is being tested. We added these details in Section 4.3 and more information in Section 4.2.1 of the supplementary material. For multi-domain plan generation using a single model, we developed a single Plansformer model trained on all four domains and tested on these four domains. The test bed on which this single Plansformer model is tested is the same as the test bed used by Plansformer-bw, Plansformer-hn, Plansformer-gr, and Plansformer-dl, i.e., each domain's test bed consists of 3000 problem instances. The results obtained are as follows -
>
> | Test Domain | Valid Plans | Cost-Optimal Plans | Invalid Plans                       |
> |-------------|-------------|--------------------|-------------------------------------|
> | **bw**      | 95.83%      | 93.75%             | Failed = 4.17%, Incomplete = 0%     |
> | **hn**      | 79.25%      | 76.72%             | Failed = 2.34%, Incomplete = 18.41% |
> | **gr**      | 78.44%      | 50.61%             | Failed = 15.03%, Incomplete = 6.53% |
> | **dl**      | 63.03%      | 55.25%             | Failed = 2.81%, Incomplete = 34.16% |
>
> The single Plansformer trained on all models outperforms Plansformer-bw by 5.79% (in terms of valid plans) and has around a 4.5% to 13.5% decrease in valid plans for the other three domains. These details have been added in Section 4.2.2 of the supplementary material.
>
> **References**
>
> [1] Valmeekam, K., Olmo, A., Sreedharan, S., & Kambhampati, S. (2022). Large Language Models Still Can't Plan (A Benchmark for LLMs on Planning and Reasoning about Change). arXiv preprint arXiv:2206.10498.
>
> [2] Helmert, M., & Domshlak, C. (2011). Lm-cut: Optimal planning with the landmark-cut heuristic. Seventh international planning competition (IPC 2011), deterministic part, 103-105.

---

> > ### Comment · Reviewer_X39q · 2022-11-22
> > **This is all more clear**
> >
> >
> > I think the additional comments and experiments has improved my perception of the work. I'm increasing my scores. I don't think the paper would be accepted, but I think the paper is in good shape.
> >
> > I still dislike that the model Plansformers-X indicate the data tested. I think an extra column should be added in Table 1 and Table 2.
> > Figure 5 is more clear because the testing data is explicit.
> >
> > For the record, I think it'd be good to test Plansformers in bigger instances.
> > Otherwise, we have no idea how close it gets to a classical planner.  The other models would, of course, won't scale up.
> >
> > I think the current manuscript misses an opportunity to emphasize how metrics used in dialogue and text generation are so informative for planning. That should be in the abstract and in the introduction.

---

> > > ### Author Response · Authors · 2022-11-22
> > > **Thank you!**
> > >
> > > We appreciate your revision of scores and we thank you for your helpful reviews once again!

---

> > ### Comment · Reviewer_abxA · 2022-11-24
> > **Thanks**
> >
> > I think Fig.5d is very helpful and makes the claim on transfer learning much stronger. Thanks a lot for doing that.
> >
> > With these added experiments, I think the claims are better supported, and the manuscript is much clearer now. I've raised my score.
> >
> > If you get a chance (not requesting for reviewing purpose), it may be helpful to generate your own test set with object names randomized on all domains. just to have the experiments in a more controlled setting. Also, the result in Supp 4.2 (testing with different object numbers) is important and may be worth mentioning in the main text.

---

### Comment · Reviewer_X39q · 2022-11-23
**Significance of the problem**

I wonder if the authors could comment on how difficult is the tasks. For instance, what’s the performance we should expect from

- solving them by blind search
- how hard it would be to train RL models for them? In this case, the proper comparison depends on the domain. One thing is a model for the domain blocks, something else is a generalist model.

Another question is about the consequences of this work.
I don’t think commenting on specific work on reasoning using LLM would not be illuminating because any synthesis would be disputed.
Instead, could you summarize your points or elaborrate on how could this be used in other domains beyond the planning benchmarks.

---

### Comment · Reviewer_X39q · 2022-11-24
**Randomization**

Dear authors, I think randomization is not an issue.
The main reason is that the diversity of the problems is even harder.

Would you summarize the issue, the concerns, and your answer?

Please do your best on summarizing the outstanding issues.

I think this would help us settle the differences.

Feel free to summarize any outstanding issue or angle.

---

### Author Response · Authors · 2022-11-28
**Addressing additional questions**

***On object randomization =>*** We have performed additional experiments to answer the questions related to object randomization. The experimental setup involved creating randomized object names for every instance in four datasets (**bw**, **hn**, **gr**, and **dl**). We have created two  different types of random prompts:
  - Prompt 1 - consists of only single-digit numbers as object names.
  - Prompt 2 - consists of an alphanumeric string of length 2.
Two prompt types used for the same problem instance (from **bw**) are as follows:

| **Prompt** |                                                                                                                                                                                                                                                                                   **Problem Instance**                                                                                                                                                                                                                                                                                  |                                                                                   **Ground-truth Plan  (generated by FastDownward)**                                                                                  |
|:----------:|:---------------------------------------------------------------------------------------------------------------------------------------------------------------------------------------------------------------------------------------------------------------------------------------------------------------------------------------------------------------------------------------------------------------------------------------------------------------------------------------------------------------------------------------------------------------------------------------:|:---------------------------------------------------------------------------------------------------------------------------------------------------------------------------------------------------------------------:|
| Prompt 1   | <GOAL>on 1 7, on 8 2, ontable 7, on 2 1, on 4 8, clear 4<INIT>handempty, on 1 7, on 8 1, on 7 4, on 2 8, clear 2, ontable 4<ACTION> pick-up <PRE> clear x, ontable x, handempty <EFFECT> not ontable x, not clear x, not handempty, holding x <ACTION> put-down <PRE> holding x <EFFECT> not holding x, clear x, handempty, ontable x <ACTION> stack <PRE> holding x, clear y <EFFECT> not holding x, not clear y, clear x, handempty, on x y <ACTION> unstack <PRE> on x y, clear x, handempty <EFFECT> holding x, clear y, not clear x, not handempty, not on x y                     | unstack 2 8, put-down 2, unstack 8 1, put-down 8, unstack 1 7, stack 1 8, unstack 7 4, put-down 7, unstack 1 8, stack 1 7, pick-up 2, stack 2 1, pick-up 8, stack 8 2, pick-up 4, stack 4 8                           |
| Prompt 2   | <GOAL>on e4 rq, on 1j x2, ontable rq, on x2 e4, on db 1j, clear db<INIT>handempty, on e4 rq, on 1j e4, on rq db, on x2 1j, clear x2, ontable db<ACTION> pick-up <PRE> clear x, ontable x, handempty <EFFECT> not ontable x, not clear x, not handempty, holding x <ACTION> put-down <PRE> holding x <EFFECT> not holding x, clear x, handempty, ontable x <ACTION> stack <PRE> holding x, clear y <EFFECT> not holding x, not clear y, clear x, handempty, on x y <ACTION> unstack <PRE> on x y, clear x, handempty <EFFECT> holding x, clear y, not clear x, not handempty, not on x y | unstack x2 1j, put-down x2, unstack 1j e4, put-down 1j, unstack e4 rq, stack e4 1j, unstack rq db, put-down rq, unstack e4 1j, stack e4 rq, pick-up x2, stack x2 e4, pick-up 1j, stack 1j x2, pick-up db, stack db 1j |

We trained randomized Plansformer models (plansformer trained on object randomized dataset), one each on different prompt types for the blocksworld domain - written as Plansformer-bw(random)-Prompt[x]. We then tested the randomized models on the dataset released in [1]. The object names in [1] are single lettered alphabets. The performance of different models on 500 **bw** problem instances from [1] are shown in the table below:

|            **Model**           | **Valid Plans  (out of 500)** |
|:------------------------------:|:-----------------------------:|
| GPT-3 + Prompting [1]          | 0.60%                         |
| Plansformer-bw                 | 66.00%                        |
| Plansformer-bw(random)-Prompt1 | 67.40%                        |
| Plansformer-bw(random)-Prompt2 | 27.52%                        |

---

> ### Author Response · Authors · 2022-11-28
> **Addressing additional questions (contd.)**
>
> ***On object randomization (contd.) =>*** We observe that the model trained on a randomized **bw** dataset according to Prompt 1 achieves 67.40% valid plans, followed by Plansformer-bw (original model trained on the non-randomized dataset with objects named b1, b2, and so on). We find  that the performance drops as we include more alphabets in the object names. From this experimentation, we also observe that the models perform better when the object names look different  than the action names. On the other hand , if we follow a simple technique of mapping object names [5] from the new test set to the object names similar to the train set and generated a plan with this mapped problem instance as input to any Plansformer-bw model (randomized or otherwise), we were able to achieve around 80% valid plans (Plansformer-bw(random)-Prompt1 gave 81.97% valid plans, Plansformer-bw(random)-Prompt2 gave 80.29% valid plans, and Plansformer-bw gave 82.56% valid plans). Overall, the object names play a role in the generation of plans. We would like to report our findings on object randomization from here in the revised version of the paper.
>
> For the rest of the  experiments on object randomization, we consider Prompt 2 since we would like to understand how Plansformer performs in the worst-case setting  for plan generation.
>
> Next, we trained Plansformer on the randomized datasets for  each planning domain with Prompt 2. The naming convention used for these models is as follows - Plansformer-<domain_name>(random)-Prompt2. Plansformer-<domain_name> implies  the base model from  the paper. We have tested the randomized models and base models on both randomized and original validation datasets. The table below shows the performance of these models:
>
> |            **Model**           |    **Train Domain**   |       **Test Domain**      | **Valid Plans  (out of 3000)** | **Cost-Optimal Plans  (out of 3000)** |   **Invalid Plans  (out of 3000)**   |
> |:------------------------------:|:---------------------:|:--------------------------:|:------------------------------:|:-------------------------------------:|:------------------------------------:|
> | Plansformer-bw(random)-Prompt2 | Randomized bw-Prompt2 | Randomized bw-Prompt2-test | 89.83%                         | 86.42%                                | Failed = 10.17%, Incomplete = 0%     |
> | Plansformer-hn(random)-Prompt2 | Randomized hn-Prompt2 | Randomized hn-Prompt2-test | 69.56%                         | 66.78%                                | Failed = 24.97%, Incomplete = 5.47%  |
> | Plansformer-gr(random)-Prompt2 | Randomized gr-Prompt2 | Randomized gr-Prompt2-test | 75.89%                         | 70.56%                                | Failed = 22.47%, Incomplete = 1.64%  |
> | Plansformer-dl(random)-Prompt2 | Randomized dl-Prompt2 | Randomized dl-Prompt2-test | 67.86%                         | 63.08%                                | Failed = 19.25%, Incomplete = 12.89% |
> | Plansformer-bw(random)-Prompt2 | Randomized bw-Prompt2 | bw-test                    | 48.22%                         | 45.22%                                | Failed = 45.17%, Incomplete = 6.61%  |
> | Plansformer-hn(random)-Prompt2 | Randomized hn-Prompt2 | hn-test                    | 10.42%                         | 8.72%                                 | Failed = 69.33%, Incomplete = 20.25% |
> | Plansformer-gr(random)-Prompt2 | Randomized gr-Prompt2 | gr-test                    | 3.06%                          | 2.86%                                 | Failed = 53.86%, Incomplete = 43.08% |
> | Plansformer-dl(random)-Prompt2 | Randomized dl-Prompt2 | dl-test                    | 2.69%                          | 1.04%                                 | Failed = 33.94%, Incomplete = 63.37% |
> | Plansformer-bw                 | bw                    | Randomized bw-Prompt2-test | 70.69%                         | 68.14%                                | Failed = 29.25%, Incomplete = 0.06%  |
> | Plansformer-hn                 | hn                    | Randomized hn-Prompt2-test | 34.58%                         | 34.03%                                | Failed = 43.72%, Incomplete = 21.7%  |
> | Plansformer-gr                 | gr                    | Randomized gr-Prompt2-test | 28.39%                         | 27.31%                                | Failed = 2.08%, Incomplete = 69.53%  |
> | Plansformer-dl                 | dl                    | Randomized dl-Prompt2-test | 20.64%                         | 20.64%                                | Failed = 0.06%, Incomplete = 79.30%  |
>
> We see that the randomized Plansformer models generate lesser valid plans than the base models, with a maximum drop in performance of 15.41% in the Hanoi domain (Plansformer-hn produces 84.97% valid plans on hn vs 69.56% valid plans with randomized train and test domains). In general, the base models show a better performance than the randomized models, owing to consistent object naming convention followed during training.

---

> > ### Author Response · Authors · 2022-11-28
> > **Addressing additional questions (contd.)**
> >
> > ***On training a model with a larger dataset by including problem instances with randomized object names =>*** To create a larger dataset, as per the Reviewer T1dJ’s suggestion, in addition to the original 14,400 problem instances shown in our paper for blocksworld, we included the problem instances based on Randomized bw-Prompt1, and Randomized bw-Prompt2.. The new **bw** training set for plansformer now consists of 43,200 problem instances. The table below shows the performance of Plansformer  trained on this larger dataset and tested on various **bw** test sets:
> >
> > |                 **Test Domain**                | **Valid Plans** |
> > |:----------------------------------------------:|:---------------:|
> > | bw-test+Randomized bw-[Prompt1 & Prompt2]-test | 99.06%          |
> > | bw-test                                        | 97.77%          |
> > | Randomized bw-Prompt1-test                     | 98.41%          |
> > | Randomized bw-Prompt2-test                     | 96.25%          |
> >
> > As pointed out by the reviewer, the new model trained on 43k problem instances  outperforms other existing Plansformer-bw variants by a fair margin. The original Plansformer-bw base model generated 90.04% valid plans (tested on non randomized bw test-bed), whereas the new model trained on the larger dataset obtained 97.77% on the same test bed (second row in the table above). The result is quite interesting as the new model trained on 3 different variants of the same problem instance gave a performance boost  of 7%, providing us with alternate ways for the language model on planning domain to learn better.
> >
> > ***On complexity and diversity of the domains =>*** As requested by Reviewer X39q, we have performed a blind search on the test sets of the four domains (3000 problem instances per domain). The table below captures the results obtained by A* + LM-Cut (informed search) and Breadth-First Search (uninformed/blind search):
> >
> > | Domains | A* + LM-Cut (Generated States & Evaluated States) | Blind Search (Generated States & Evaluated States) |
> > |---------|---------------------------------------------------|----------------------------------------------------|
> > | bw      | 51 & 35                                           | 707 & 334                                          |
> > | hn      | 141 & 55                                          | 201 & 76                                           |
> > | gr      | 33520 & 1347                                      | 3795846 & 381627                                   |
> > | dl      | 188 & 131                                         | 568337 & 110498                                    |
> >
> > The results reported above are  averaged over all 3000 problem instances per domain. Generated States refer to the total number of states obtained for the given problem instance, whereas, the evaluated states refer to the number of traversed generated states to arrive at the goal state. The results obtained by blind search provide an insight  about the complexity of the domains (with hn and bw on the easier side vs dl and gr on the harder side). Both the search strategies solved 3000 problems, but the blind search took an average of 214 seconds to solve the test bed, whereas A* + LM-Cut took 14.72 seconds.
> >
> > All the considered planning domains have a diverse set of state-action space. Because of the diversity of these domains, we observed that in “multi-domain plan generation using a single model” setup from Section 4.2.2 in supplementary material, single model trained on problem instances from all four domains has an unfair advantage for easier domains such as bw (5.79% increase in terms of valid plans on  Plansformer-bw vs 90.04% in Table 2) compared to performance drop for harder domains such as gr and dl (around a 4.5% to 13.5% decrease in valid plans).
> >
> > ***On training RL models for the domains =>*** Regarding the question on RL agents, we can provide an estimate of the search space. For a plan of length L and actions m that can take n maximum parameters, the search space size is (L) ^ (m x n x k), where k is the number of objects in the problem. So, it increases exponentially with the number of actions, parameters for the actions, and the number of objects. Additionally, the long-horizon goal-based problems found in classical planning produce sparse rewards for RL, posing a problem in the difficulty for the direct application of RL. Training RL models for such domains is an exciting area of research, and [2, 3] gives a sense of the issues involved in the application of RL in these domains.

---

> > > ### Author Response · Authors · 2022-11-28
> > > **Addressing additional questions (contd.)**
> > >
> > > ***Consequence of this work =>***  In this work, we investigate a way to use language models trained on code for generating symbolic plans. Automated Planning has multiple planning types (for example, classical planning, epistemic planning). However, building a single general model that can adapt to various planning types at scale is an ongoing research area. Our model, Plansformer, can be used as a building block for general and adaptive planners. Apart from generating valid plans for automated planning domains, we are currently working on using Plansformer with an existing automated planner (such as FastDownward) in the context of a cognitive architecture inspired by the thinking fast and slow theory [4]. In addition, Plansformer may also be used as an instruction-following framework in robot navigation, unmanned vehicles, embodied artificial intelligence, etc [5] and potentially as an automated planner in storytelling/dialogue generation [6, 7]. One significant advantage of using Plansformer (possibly in combination with an automated planner) in these domains is to solve complex problems in constant time, as a current automated planner (such as FastDownward) is incapable of solving complex problems without reaching time or memory exhaustion.
> > >
> > > ***Other outstanding issues, concerns and possible solutions =>*** In our response so far, we have addressed the outstanding issues raised by the reviewers regarding -
> > >   - object randomization where the object names are randomized during training and testing to evaluate the performance of Plansformer.
> > >   - complexity & diversity of the domains where we have discussed the difficulty of the domains based on the generated/evaluated states and diverse set of search space.
> > >   - single model for multi-domain setup where we build a single model trained on problem instances from all the domains.
> > >   - transfer learning on different domains where we demonstrated Plansformer’s ability to generate plans when tested on problems consisting of plan length longer than that of the train set.
> > >
> > > We hope that we have addressed the reviewers' questions to the best of our abilities and that they find these new experimental results supportive of our work.
> > >
> > > **References:**
> > >
> > > [1] Valmeekam, K., Olmo, A., Sreedharan, S., & Kambhampati, S. (2022). Large Language Models Still Can't Plan (A Benchmark for LLMs on Planning and Reasoning about Change). arXiv preprint arXiv:2206.10498.
> > >
> > > [2] Agostinelli, F., McAleer, S., Shmakov, A., Fox, R., Valtorta, M., Srivastava, B., & Baldi, P. (2021). Obtaining Approximately Admissible Heuristic Functions through Deep Reinforcement Learning and A* Search. In Bridging the Gap Between AI Planning and Reinforcement Learning Workshop at ICAPS (Vol. 2021).
> > >
> > > [3] Toyer, S., Thiébaux, S., Trevizan, F., & Xie, L. (2020). Asnets: Deep learning for generalised planning. Journal of Artificial Intelligence Research, 68, 1-68.
> > >
> > > [4] https://www.nasa.gov/content/planning-scheduling
> > >
> > > [5] Huang, W., Abbeel, P., Pathak, D., & Mordatch, I. (2022). Language models as zero-shot planners: Extracting actionable knowledge for embodied agents. arXiv preprint arXiv:2201.07207.
> > >
> > > [6] Yao, L., Peng, N., Weischedel, R., Knight, K., Zhao, D., & Yan, R. (2019, July). Plan-and-write: Towards better automatic storytelling. In Proceedings of the AAAI Conference on Artificial Intelligence (Vol. 33, No. 01, pp. 7378-7385).
> > >
> > > [7] Pallagani, V., & Srivastava, B. (2021). A Generic Dialog Agent for Information Retrieval Based on Automated Planning Within a Reinforcement Learning Platform. Bridging the Gap Between AI Planning and Reinforcement Learning (PRL).

---

> > ### Comment · Reviewer_T1dJ · 2022-11-30
> > **Thank you**
> >
> > I appreciate the effort taken by the authors to quantify the effects of object name randomization.
> >
> > This table and the paper's Table 2 tell a complex story. When train and test use the *same* naming scheme (either original or fully randomized), then the impact of object names on valid plans is highly task-specific, ranging from a very small drop (0.21%) on Blocks World, to a significant drop (15.41%) on Hanoi. When train and test *do not* employ the same object naming scheme, validation performance falls dramatically, from 76.56% to 2.69% in the worst case.
> >
> > Have I characterized these results correctly?

---

> > > ### Author Response · Authors · 2022-11-30
> > > **In response to Reviewer T1dJ's observations**
> > >
> > > The result has to be interpreted in the context of how actions and objects are named in planning domains (e.g., in International Planning Competition) - the variables are parameterized in the action names with a unique prefix or suffix. The action names are not mixed with object names (i.e., do not share same vocabulary) as they are intended to be understood by people. Our results show that randomization of variables can have a performance impact ranging from maintaining the performance to degrading it based on the difficulty/complexity of the domain as shown. Since an LLM learns over its inputs, such a performance is not surprising. One easy way to retain performance is by mapping the unique parameter names in test problems to internal naming scheme (used during training) that does not degrade the performance.

---

> > > > ### Comment · Reviewer_T1dJ · 2022-11-30
> > > > **Object name randomization**
> > > >
> > > > Thanks for the extra info.
> > > >
> > > > Your new results seem to be about randomization of *object* names rather than *action* names. In this context, do you see anything incorrect about my short summary of the results?

---

> > > > > ### Author Response · Authors · 2022-12-01
> > > > > **Object name randomization**
> > > > >
> > > > > Name of an instantiated action in an executable plan is a combination of action names and object names (like: unstack x2 1j).

---

> > > > > > ### Comment · Reviewer_T1dJ · 2022-12-01
> > > > > > **Object name randomization**
> > > > > >
> > > > > > Ok, let's use that terminology:  Each **instantiated action** is a combination of an **action name** with one or more **object names**. In the examples that you gave at the start of this thread, only the object names were randomized, unless I missed something. Then you provided a table of results obtained by randomizing the object names for either train or test or both. Do you see anything incorrect about the following summary of some highlights from that table?
> > > > > >
> > > > > > *This table and the paper's Table 2 tell a complex story. When train and test use the **same** object naming scheme (either original or fully randomized), then the impact of object names on valid plans is highly task-specific, ranging from a very small drop (0.21%) on Blocks World, to a significant drop (15.41%) on Hanoi. But when train and test **do not** employ the same object naming scheme, validation performance falls dramatically, from 76.56% to 2.69% in the worst case.*
> > > > > >
> > > > > > Thank you.

---

### Comment · Reviewer_X39q · 2022-12-01
**[post discussion reflection] What's the contribution?**

Dear authors,
as the discussion period is almost over, please don't bother answering this thread.
I leave it here, just in case the authors find it useful.

----

Dear authors,

After discussing with other reviewers, I think the contribution of the paper is still confusing in the paper.

The current manuscript sometimes tries to show that transformers would be able to plan in many circumstances, including variations of the object names.

I think the original point is an empirical demonstration that transformers could plan, reporting what methods work better.
That doesn't mean that any object naming should work.

Perhaps it'd be simpler and more transparent to report how to make transformers plan in these non-trivial domains.
Part of the how is a strategy for object naming is akin to prompt design.
I'm not sure prompt design would make it into AI textbooks in 20 years but, for know, it's a valid contribution.
Showing how small changes affect performance is useful, but that's not the main point.

The significance of the work I think has been justified enough: pre-trained models are not performing well, and the paper presents a way to do it.

Revisiting the randomization, besides my point above, I suggest making it more clear that this does not imply a general strategy for any kind of text generation that involves planning-like problems. It's possible that the strategy used for randomization here might not be feasible in other domains. Furthermore, getting plans of high quality might be very expensive or even impossible in some domains –unless a symbolic model is available, that would in turn enable the object naming strategy here.

We cannot get another version of the paper, so perhaps the authors can answer this two questions:
* What's the contribution, and what's not a contribution?
* What would you tune down or even remove from the introduction and the discussion of results to make the paper more clear around the contribution?
* How would you use that space?

---

> ### Comment · Reviewer_X39q · 2022-12-01
> **Are the transformers memorizing?**
>
> Dear authors,
> as the discussion period is almost over, please don't bother answering this thread.
> I leave it here, just in case the authors find it useful.
>
> ----
>
> I said somewhere else that I find it very unlikely that a transformer model is latching on correlations just because there are so many instances. But I'm familiar with planning, so this might not be obvious for many readers.
> Perhaps a short discussion on coverage would be useful. What fragment of the spaces of the possible plans is being covered?
> What variables could produce bias and was that remediated?
> If there is enough coverage of combinations of problems –not at symbolic level, but as input sequences to the transformer–, then the issue of possible memorization and randomization might be less important.
>
> If the authors plan to respond to this thread, I suggest summarizing what would be the changes to the paper.
> There are many new experiments and discussions, and it's hard to keep track.

---

> > ### Author Response · Authors · 2022-12-03
> > **On Memorization**
> >
> > We do not think memorization is happening in our approach. Regarding the plans' coverage in the dataset, we have created an extensive set of problem instances for each domain (~200,000 problem instances per domain) that cover various object configurations for the respective domain. For example, in blocksworld, we consider 2 to 5 block configurations and generate all possible problem instances and corresponding plans (plans are generated using FastDownward). Once the plans are obtained, we use the set function to remove any duplicate plans and then select 18,000 problem instances of varying plan lengths with different initial goal conditions and the number of objects to obtain the dataset for Plansformer. A similar approach is followed with all domains to prevent any data correlations. We want to add the complete details on dataset creation in the supplementary material, along with qualitative results showing no correlation between the instances present in the dataset. We had a longer discussion on the theme on November 20, 2022.

---

### Author Response · Authors · 2022-12-03
**Our Contribution**

As requested, we are summarizing the main contributions. They are as stated in the submitted version:
  1. We introduce Plansformer; an LLM fine-tuned on planning problems and capable of generating plans with favorable behavior in terms of correctness and length with minimal knowledge-engineering efforts for four different planning domains (Hanoi, Gripper, Driverlog, Blocksworld).  For one configuration of Plansformer - Towers of Hanoi - a puzzle-solving domain, we achieved 97% valid plans, out of which 95% are optimal. This is in contrast to negative results with GPT-3 reported in literature where they found 0.6% valid plans in blocksworld; we were able to generate 66% valid plans in the best case.
  2. We also demonstrate the generalization ability of Plansformer in solving different planning domains (Hanoi, Gripper, Driverlog, Blocksworld) with varying complexities, owing to the transfer learning abilities of LLMs.

During the review period, with the guidance of our reviewers, for whom we are indebted, we conducted additional experiments which shed light on Plansformer's abilities. One of them, in particular, was Plansformer's capability in plan generation when the plan length of the test set is considerably larger than that of the train set (currently, results are present in Sec 4.2 of supplementary).

So, in the revised paper, beyond the latest content, we will:
  - Reorganize the discussion on experimental results in the main paper to include Section 4.2 from the supplementary.
  - Describe experiments conducted with different prompts in the supplementary material.

---

### Author Response · Authors · 2022-12-12
**Note of Thanks!**

We thank all the reviewers for sharing their valuable feedback during the rebuttal period to help enhance the paper. We hope we have addressed the feedback to the best of our abilities.

---

### Decision · Program_Chairs · 2023-01-20

**Decision:**

Reject

**Justification For Why Not Higher Score:**

The paper is at best an interesting exploration of how one might fine-tune LLMs for use in classical AI planning.  It does not constitute a substantial technical advance.

**Justification For Why Not Lower Score:**

N/A

**Metareview: Summary, Strengths And Weaknesses:**

This paper provides a strategy for solving classical AI planning problems by fine-tuning a code-oriented large language model.    The model is fine-tuned using data generated automatically by calling the planner on the problem domain specification.  It is then applied to different problem instances, including larger ones, drawn from the same formal domain.   Additional results on transfer are included, as well as experimentation (done at the behest of the reviewers) on the effect of randomizing object names.

Viewed in terms of the effectiveness of the resulting artifact as a planning algorithm, the results are mixed:
- It solves instances of relatively simple domains with high reliability
- It is less reliable (75%) on more difficult domains
- It generates solutions much more quickly than a planner
- It's not clear what the amortized cost of the fine-tuning is

Despite the lack of reliability as a planner, the contribution raises interesting points:
- It teaches us about the strengths and weaknesses of LLMs for planning
- It explores fine-tuning (rather than prompt conditioning, which had been the strategy of previous papers) and finds it to be helpful

The paper was a bit muddled in terms of
- exactly what points it was trying to make
- presentation of results (it is critical to make clear, for every single number, what data it is being trained and tested on)
But the authors did clarify substantially in a constructive dialog with the reviewers

A smaller point is that I don't think anyone found the Bleu-score results motivating---the big question is whether the plans are valid.  Those results could be moved to the appendix in favor of more results about actual planning performance.

**Summary Of Ac-Reviewer Meeting:**

The reviewers raised many points about the details and significance of the results, in terms of randomizing object names, ability to generalize vs memorize, ability to perform well on instances that are larger than the training instances, etc.

The authors worked hard to address these concerns, but to some reviewers it felt like the conclusions were not stated crisply and very clearly substantiated.  Other reviewers felt the work was interesting and timely and worth publication.